# Utilization of Neighbor Information for Generalized Category Discovery and Image Clustering

## Abstract

We propose to bridge the gap between semi-supervised and unsupervised image recognition with a flexible method that performs well for both generalized category discovery (GCD) and image clustering. Despite the overlap in motivation between these tasks, the methods themselves are restricted to a single task – GCD methods are reliant on the labeled portion of the data, and deep image clustering methods have no built-in way to leverage the labels efficiently. We connect the two regimes with an innovative approach that **U**tilizes **N**eighbor **I**nformation for **C**lassification (**UNIC**) both in the unsupervised (clustering) and semisupervised (GCD) setting. State-of-the-art clustering methods already rely heavily on nearest neighbors. We improve on their results substantially in two parts, first with a sampling and cleaning strategy where we identify accurate positive and negative neighbors, and secondly by finetuning the backbone with clustering losses computed by sampling both types of neighbors. We then adapt this pipeline to GCD by utilizing the labelled images as ground truth neighbors. Our method yields state-of-the-art results for both clustering (+3% ImageNet-100, Imagenet-200) and GCD (+0.8% ImageNet-100, +5% CUB-200).

## 1 Introduction

Image recognition has long been treated as a fundamental task in computer vision. One very popular setting is supervised image classification, where each image has a single ground truth label, and a model learns to predict labels by training with image-label pairs (Deng et al., 2009). As work in the fully-supervised setting plateaus, interest in the field has shifted to settings more reminiscent of the real world, where access to labeled data is more limited, or even completely absent. So, many different recognition tasks have evolved based on the presence or absence, fraction of presence, quality, granularity, and type of labels (Koch et al., 2015; Bendale & Boult, 2015; Snell et al., 2017; Vinyals et al., 2017; Suri et al., 2023). At the extreme, unsupervised classification (Gansbeke et al., 2020), or deep image clustering, deals with the case where none of the labels are disclosed during the training step. Recently, generalized category discovery (GCD) (Vaze et al., 2022) has been proposed, such that some images from some classes are labelled, and some classes are completely unlabelled.

We propose a method, UNIC (pronounce "you-neek"), which can solve both clustering and GCD, as shown in Figure 1. When labels are unavailable, UNIC can operate as a state-of-the-art deep image clustering pipeline. Where labels are available, UNIC can leverage them for SOTA performance on GCD. We accomplish this by designing UNIC to exploit neighbor information in the feature space of high quality unsupervised image representation models. For every image, we mine both "positive" (belongs to the same class) and "negative" (belongs to a different class) neighbors. We then finetune the unsupervised model end-to-end with losses that pull the positive neighbors together, and push the negative neighbors apart. This adapts quite naturally to GCD, since we can use the labels to guarantee 100% accuracy for a portion of the positive and negative neighbors.

We find that with the unsupervised DINO (Caron et al., 2021) ViT (Dosovitskiy et al., 2021) backbone, we can reliably extract both positive and negative neighbors. The success of our method is fully dependent on the quality of these neighbors. So, we carefully analyze the representation space

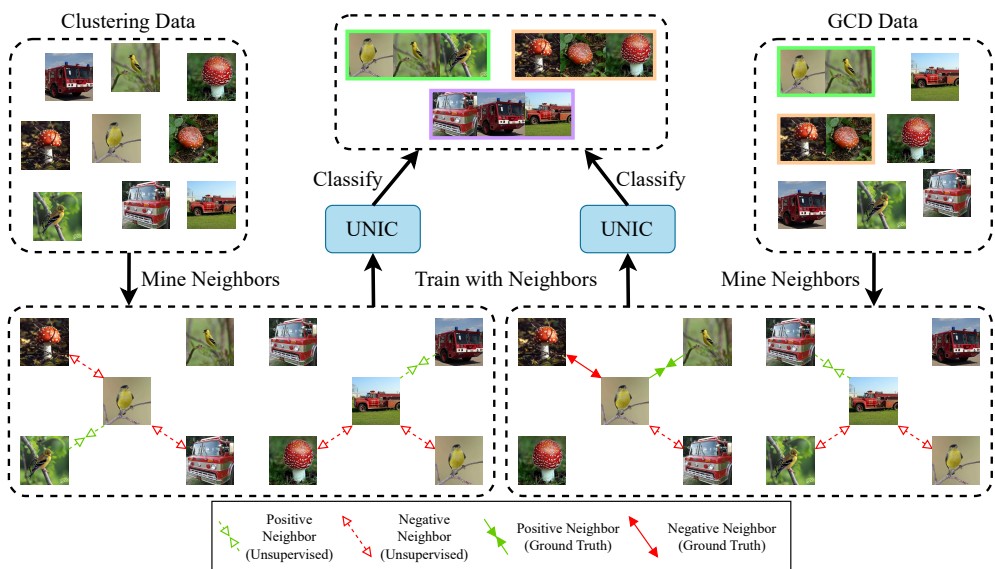

Figure 1: **Unifying Clustering and GCD.** We observe that the goals of image clustering and generalized category discovery (GCD) are identical, they only differ slightly in terms of supervision (top). Therefore, we propose a clustering approach based on mining of positive and negative neighbors, which belong to the same class as an anchor and a different class, respectively (bottom left). We can extend this approach for GCD by using the ground truth labels for "perfect" neighbors (bottom right).

and propose a novel nearest-neighbor cleaning strategy for "positive" neighbors. Intuitively, images at the decision boundary will have noisy neighbors (because the neighbors belong to different classes) while images near the centroid of classes will have cleaner neighbors. We identify the union size of second-order neighbors (the neighbors of neighbors) as a proxy for this purity measure and discard others.

We improve on prior clustering work that relies on a three-stage pipeline (Gansbeke et al., 2020) with our two-stage pipeline, where aside from the initial representation learning, our clustering is learned in one step. After neighbor mining, we finetune the model end-to-end. Class assignment is learned with a clustering loss. Meanwhile, we use a maximum entropy loss to ensure that the clustering does not collapse to a trivial solution. Due to the strength of the DINO backbone itself, the influence of the negative neighbors, and the fact that we are able to successfully finetune, we find that the self-labeling step is not necessary, and gives diminishing returns.

Once we establish the good clustering properties of the method, we apply it to the GCD task. GCD methods typically rely on some clustering for initialization (Vaze et al., 2022; Pu et al., 2023). However, we find such initialization totally unnecessary as we can instead learn a partially-supervised clustering end-to-end. That is, we are able to treat the labelled images as a special case where we simply supervise their neighbor mining.Strikingly, this results in much better performance on the labeled classes for other method compared to others, while also having competitive results in unlabeled classes.

We summarize our contributions as follows:

- We propose a novel neighbor mining strategy where we improve the accuracy of positive neighbors via cleaning and introduce the idea of negative neighbors for image clustering.

- We formulate a general pipeline which can be trained end-to-end for image clustering and, with minimal adaptation, for the semi-supervised GCD task as well.

- We achieve SOTA on both tasks, unifying classification for unsupervised and semi-supervised methods, with consistent gains over most datasets in both GCD and Clustering.

## 2 RELATED WORK

### 2.1 UNSUPERVISED REPRESENTATION LEARNING

Unsupervised representation learning refers to the process of training deep neural networks to extract information from images without labels. Some early approaches rely on a host of diverse pretext tasks – patch prediction (Doersch et al., 2015; Gupta et al., 2020), jigsaw solving (Noroozi & Favaro, 2016), colorization (Larsson et al., 2016; Zhang et al., 2016), inpainting (Pathak et al., 2016), rotation prediction (Gidaris et al., 2018), image generation (Donahue & Simonyan, 2019), etc. (Doersch & Zisserman, 2017). Contrastive (Wu et al., 2018; Chen et al., 2020b;c; Misra & Maaten, 2020; Bachman et al., 2019; Chen et al., 2020a; He et al., 2019; Hjelm et al., 2018; Henaff, 2020; Oord et al., 2018; Tian et al., 2020; Ye et al., 2019; Zbontar et al., 2021; Chen & He, 2021) and clustering (Caron et al., 2018; Asano et al., 2019; Caron et al., 2019; 2020; 2021; Li et al., 2021) methods have largely superseded these. More recent approaches rely on image reconstruction (He et al., 2021; Assran et al., 2022; Zhou et al., 2022; Huang et al., 2022; Bao et al., 2022; Mishra et al., 2022) and generation (Mukhopadhyay et al., 2023; Hudson et al., 2023; Li et al., 2022). Currently these clustering, contrastive, and reconstruction-based approaches are all among the state-of-the-art. In our work, since we do deep clustering (an unsupervised problem), we need backbones which do not require labels, and we primarily use a ViT-B (Dosovitskiy et al., 2021) trained with DINO (Caron et al., 2021) due to the nice properties of its embeddings with respect to nearest neighbors.

### 2.2 DEEP CLUSTERING

Deep clustering aims to learn groups of images from unlabelled data, where the images in a given grouping (cluster) have similar visual semantics. A naive approach would use kmeans on top of some embeddings extracted by a model trained with unsupervised representation learning, and indeed, this task has been used as a benchmark for the quality of such representations (Gwilliam & Shrivastava, 2022). The work in this area can be primarily divided between single-stage and multi-stage methods, as in (Adaloglou et al., 2023). Some of these target the clustering problem directly (Chang et al., 2017a), but many of the single-stage methods use clustering primarily as a mechanism for learning good image representations (Caron et al., 2019; 2020; Asano et al., 2019). Indeed, these methods are only single-stage in the sense they have one learning stage, and they often still rely on running k-means on the final representations for the actual cluster predictions (Li et al., 2021; Huang et al., 2023).

The multi-stage pipeline is first proposed with SCAN (Gansbeke et al., 2020), which starts with a representation learning stage, followed by a stage where a clustering head is learned based on nearest neighbor mining, then a final stage where the head is further tuned with pseudo ground truths. Other methods offer deviations within this pipeline – NNM matches neighbors at both the batch and global level (Dang et al., 2021). SPICE follows similar stages, but with a heavier emphasis on pseudo-labeling (Niu et al., 2022). TSP (Zhou & Zhang, 2022b) and TEMI (Adaloglou et al., 2023) leverage ViTs to outperform the earlier methods (which mainly use ResNets (He et al., 2015)), although it is worth noting that all these methods are conceptually backbone-agnostic. Our method is conceptually aligned with SCAN and NNM except that we introduce key novelties with respect to the neighbor mining, namely with our process for cleaning positive neighbors, and our use of negative neighbors. Also, compared to many of these methods, we use a single clustering head for less expensive training.

### 2.3 GENERALIZED CATEGORY DISCOVERY

GCD is an open world image recognition task (Vaze et al., 2022). To set up some GCD task, we typically take an existing image dataset and cut the classes in half, with some known, the other unknown. We then use labels for half of the images for the known classes. The same problem is often sometimes referred to as open-world semi-supervised learning (Cao et al., 2021). The preliminary solution (also referred to as GCD) combines unsupervised contrastive learning for the unlabelled images and supervised contrastive learning for the labelled images, along with an extension of k-means to support the labelled classes (Vaze et al., 2022). DCCL (Pu et al., 2023) uses a two alternating stages in which visual concepts are discovered and then images are represented with these concepts. SimGCD (Wen et al., 2023) mitigates prediction biases in parametric classifiers with entropy

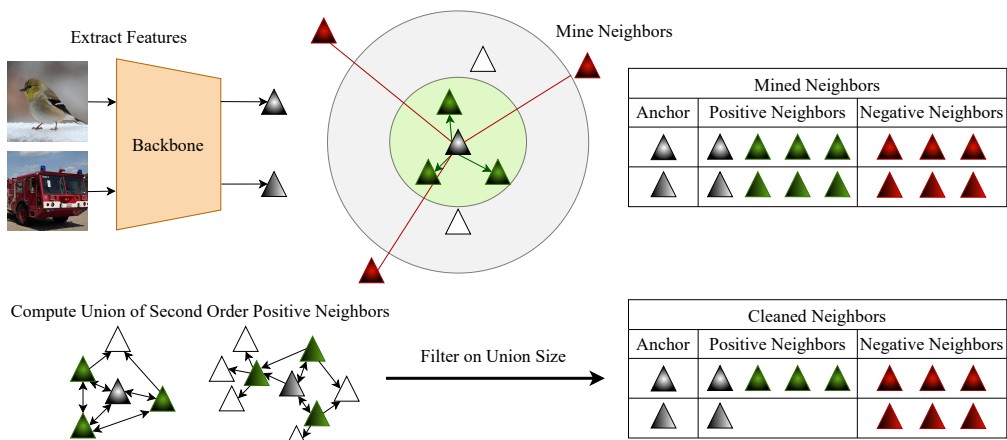

Figure 2: **Neighbor Mining** for UNIC. We extract features from a backbone, take the closest samples as "positive" neighbors, and some of the far samples for "negative" neighbors. We then prune some positive neighbors, depending on the number of mutual nearest neighbors (union of nearest neighbors of nearest neighbors).

regularization. PromptCAL (Zhang et al., 2023) utilizes graph structures for a supervisory signal. GCP (Zhao et al., 2023) uses Gaussian Mixture Models. Our method, UNIC is a parametric approach with a learnable clustering head where we reinforce connections between neighbors to learn both labelled and unlablled classes and images.

## 3 APPROACH

This section starts with a formalization of the two problem settings being addressed – image clustering and generalized category discovery. Then, we describe our pipeline in a few stages. The first stage is neighbor mining. Then, we look into designing the neural network architecture and formulating the loss function for the tasks. Finally, we describe the training process along with implementation details and hyperparameters.

### 3.1 PROBLEM SETTINGS

Consider a training set $D_t = X \times Y$ (where $X$ is the set of images and $Y$ is the set of labels) of $N = \|D_t\|$ images and a evaluation set $D_e$. $D_t$ consists of two subsets unlabelled training set $D_U = X \times Y_U$ and labelled training set $D_L = X \times Y_L$. $D_L$ is a null set for image clustering. In that case, $D_u$ is as same as $D_t$. That is, there are no labeled images.

The evaluation set $D_e$ is different for clustering and GCD settings. We use the evaluation set of the image dataset of choice (which is disjoint from the training set) for the clustering setting. However, $D_e$ is $D_U$ for the GCD setting. The objective of both clustering and GCD is to come up with a function $f$ that successfully maps $X$ to $Y$ in $D_e$. Note that $f = f_{cls} \circ f_{emb}$ for the rest of this section.

### 3.2 MINING AND CLEANING NEIGHBORS

Mining neighbors refers to embedding all the images as vectors and finding the nearest and furthest examples for every vector, as illustrated in Figure 2. We measure distances in Euclidean space. We pick the nearest $\tau_1$ examples as positive neighbors. We consider the examples that are further than the first $\tau_2$ nearest neighbors as the negative neighbors.

Consider images $x_i \in D_t$ and an embedding function $f_{emb;\theta_e}$ parameterized by $\theta_e$. The embedding vectors $v_i$ are generated as per Equation 1. Let $Q(x_i)$ be a permutation of $D_t$ sorted by the Euclidean distance of their embedding vectors to $v_i$ as per Equation 2. Let $0 < \tau_1 < \tau_2 < N$ be two integer

Figure 3: **UNIC**. We first mine neighbors (see Figure 2). We finetune the backbone with a classification head that we train without labels, using losses that encourage the model to predict the same class for positive neighbors, different classes for negative neighbors, and entropy for regularization.

parameters. We pick a set of positive neighbors $N(x_i)$ and a set of negative neighbors $\overline{N}(x_i)$ for $x_i$ as per Equation 3.

$$v_i \leftarrow f_{emb;\theta_e}(x_i); \forall x_i \in D_t. \tag{1}$$

$$Q(x_i) \leftarrow argsort_{x_j}\{\|f_{emb;\theta_e}(x_j) - f_{emb;\theta_e}(x_i)\|; \forall x_j \in D_t\} \tag{2}$$

$$N(x_i), \overline{N}(x_i) \leftarrow Q(x_i)[0:\tau_1], Q(x_i)[\tau_2:N] \tag{3}$$

It should be noted that every image $x_i$ would be the closest neighbor to itself. Cleaning neighbors refers to discarding the set of $N(x_i)$'s for whom the size of second-order neighborhoods goes beyond an integer parameter $\eta$ as described in Equation 4. Once $N(x_i)$ is discarded, the only remaining positive neighbor for $x_i$ would be itself.

$$N_{clean}(x_i) \leftarrow \begin{cases} N_i & \left|\bigcup_{x_j \in N(x_i)} N(x_j)\right| \le \eta \\ x_i & \text{otherwise} \end{cases} \tag{4}$$

## 3.3 TRAINING

Consider an image $x_i$, a positive neighbor $x_p \in N_{clean}(x_i)$ and a negative neighbor $x_n \in \overline{N}(x_i)$. These are denoted by the two red trucks and the bird in Figure 3. They are embedded into $f_{emb;\theta_e}(x_i)$, $f_{emb;\theta_e}(x_p)$ and $f_{emb;\theta_e}(x_n)$. These embedding vectors are shown by grey, green, and red triangles. Consider a classifier function $f_{cls;\theta_c}$ parameterized by $\theta_c$ that takes in an embedding vector and outputs a probability distribution over the $K$ classes. This probability distribution will be denoted by $\hat{y}_i, \hat{y}_p, \hat{y}_n \in [0,1]^K$.

$$\hat{y}_i, \hat{y}_p, \hat{y}_n \leftarrow f_{cls;\theta_c}(f_{emb;\theta_e}(x_j)); \forall x_j \in x_i, x_p, x_n \tag{5}$$

Let $< y_i, y_j >$ denote the dot product between two vectors $y_i$ and $y_j$. ~~$H(y_i)$ is the entropy of a probability distribution.~~ $H_b(a, b)$ denotes the binary cross entropy ~~of two probability distributions~~ between two probabilities $a$ and $b$. We calculate two losses $L_{pos}$ and $L_{neg}$ as per Equation 6.

$$L_{pos}(x_i, x_p) = H_b(< \hat{y}_i, \hat{y}_p >, 1.0)$$
$$L_{neg}(x_i, x_n) = H_b(< \hat{y}_i, \hat{y}_n >, 0.0) \tag{6}$$

We can calculate a loss function $L_{sim}$ for the whole dataset as per Equation 7

$$L_{sim} = \mathop{\mathbb{E}}_{x_i \in D_t}\left[\mathop{\mathbb{E}}_{x_p \in N_{clean}(x_i)}\left[L_{pos}(x_i, x_p)\right] + \mathop{\mathbb{E}}_{x_n \in \overline{N}(x_i)}\left[L_{neg}(x_i, x_n)\right]\right] \tag{7}$$

In addition, we calculate the entropy of the probability distribution of classes for all the examples in the dataset as per Equation 8. $H()$ is the entropy of a probability distribution here.

$$L_{ent} = H\left(\mathop{\mathbb{E}}_{x_i \in D_t} \left[\hat{y}_i\right]\right) \tag{8}$$

Finally, we calculate the overall loss function dataset by taking a weighted average of the two loss functions and solve the optimization problem as shown in Equation 9.

$$\theta_e^*, \theta_c^* \leftarrow \operatorname{argmin}(\alpha_{sim}L_{sim} + \alpha_{ent}L_{ent}) \tag{9}$$

The training process (optimisation problem) results in a new $f_{emb,\theta_e^*}$ which embeds images so they can be classified easily, and a new $f_{cls,\theta_c^*}$ which can classify image embeddings into classes/clusters.

### 3.4 IMPLEMENTATION DETAILS

Vit B/16 backbone is used as $f_{emb}$. A two-layer MLP is used as $f_{cls}$. $\theta_e$ is initialized with Dino (Caron et al., 2021) pretrained weights. $\theta_c$ is initialized at random. $x_i$ images undergo different transformations during training and testing times. They are always fed into the ViT as 3 colors $224 \times 224$ sized normalized tensors. The optimization problem is solved by Adam optimizer with an initial learning rate of $10^{-4}$ which is cosine annealed for 100 epochs.

We show experiments for different values of $\tau_1$ and $\eta$. We ~~pick~~ fix $\tau_2$ as 6300 for ImageNet splits, 1000 for STL and CUB datasets, and 10,000 for CIFAR10 based on heuristics.

## 4 EXPERIMENTS

Table 1: **Image Clustering Results.** We achieve SOTA on accuracy, NMI, and ARI for all datasets. [†] denotes our implementations.

| Algorithm | Backbone | STL-10 | | | ImageNet-50 | | | ImageNet-100 | | | ImageNet-200 | | |
|---|---|---|---|---|---|---|---|---|---|---|---|---|---|
| | | ACC | NMI | ARI | ACC | NMI | ARI | ACC | NMI | ARI | ACC | NMI | ARI |
| kMeans Gansbeke et al. (2020) | ResNet | 65.8 | 60.4 | 50.6 | 65.9 | 77.5 | 57.9 | 59.7 | 76.1 | 50.8 | 52.5 | 75.5 | 43.2 |
| MoCoV2[†] Chen et al. (2020c) | ResNet | 71.81 | 66.52 | 52.54 | 63.04 | 75.75 | 47.00 | 60.30 | 75.13 | 42.53 | 52.35 | 73.52 | 37.21 |
| SCAN Gansbeke et al. (2020) | ResNet | 80.9 | 69.8 | 64.6 | 76.8 | 82.2 | 66.1 | 68.9 | 80.8 | 57.6 | 58.1 | 77.2 | 47.0 |
| ProPos Huang et al. (2023) | ResNet-50 | 86.7 | 75.8 | 73.7 | - | 82.8 | 69.1 | - | 83.5 | 63.5 | - | 80.6 | 53.8 |
| DPM Ronen et al. (2022) | ResNet | 85.0 | 79.0 | 71.0 | 66.0 | 77.0 | 54.0 | - | - | - | - | - | - |
| DAC Chang et al. (2017b) | ResNet-50 | 47.0 | 36.6 | 25.7 | - | - | - | - | - | - | - | - | - |
| HCL[†] Robinson et al. (2021) | ResNet | 62.34 | 60.17 | 43.42 | - | - | - | - | - | - | - | - | - |
| MoCHi[†] Kalantidis et al. (2020) | ResNet | 69.79 | 64.00 | 50.32 | 61.88 | 73.44 | 44.92 | 57.72 | 73.44 | 42.48 | 49.16 | 70.79 | 33.38 |
| TSP Zhou & Zhang (2022b) | ViT-B/16 | 97.9 | 95.8 | 95.6 | - | - | - | - | - | - | - | - | - |
| TEMI Adaloglou et al. (2023) | ViT-B/16 | 98.5 | 96.5 | 96.8 | 80.01 | 86.10 | 70.93 | 75.05 | 85.65 | 65.45 | 73.12 | 85.20 | 62.13 |
| kMeans[†] | ViT-B/16 | 97.10 | 94.02 | 93.62 | 82.36 | 87.91 | 73.89 | 76.88 | 86.93 | 68.01 | 70.58 | 84.58 | 59.94 |
| TEMI[†] | ViT-B/16(Tune) | 98.65 | 96.70 | 97.04 | 80.12 | 86.30 | 70.64 | 76.80 | 86.41 | 67.42 | 73.69 | 85.72 | 63.16 |
| SCAN[†] | Vit-B/16 | - | - | - | 85.48 | 88.58 | 78.19 | 77.84 | 85.97 | 68.75 | 72.89[1] | 84.20 | 61.85 |
| UNIC (Ours) | ViT-B/16 | **98.75** | **96.87** | **97.25** | **90.80** | **91.81** | **84.25** | **80.84** | **88.13** | **72.70** | **75.25** | **85.93** | **64.73** |

### 4.1 EXPERIMENT SETUP

We evaluate our framework on datasets with 10, 50, 100, and 200 classes. We primarily stick with ImageNet-like datasets for their image quality and the coverage of a wide range of generic objects. It should be notes that we use a DINO ViT-B/16 backbone that is pretrained on ImageNet data without using any label information. This is crucial to have a fair comparison for unsupervised and semi-supervised tasks we perform.

We train and test UNIC on splits of ImageNet consisting of 50, 100 and 200 classes. In addition, we experiment on STL-10 dataset which consists of ImageNet-like images, but at lower resolution. For clustering, we do not utilize the labels of the training images. We follow the ImageNet-50, ImageNet-100 and ImageNet-200 splits from the recent literature to be consistent in comparison (Adaloglou et al., 2023). We use a different split for ImageNet-100 in GCD experiments to

Table 2: **Generalized Category Discovery Results** reported for all, old, and new Classes. UNIC achieves SOTA for ImageNet-100 and CUB-200.

| Algorithm | Backbone | CIFAR-10 | | | ImageNet-100 | | | CUB-200 | | |
|---|---|---|---|---|---|---|---|---|---|---|
| | | All | Old | New | All | Old | New | All | Old | New |
| kMeans Vaze et al. (2022) | DINOv1 | 83.6 | 85.7 | 82.5 | 72.2 | 75.5 | 71.3 | 34.3 | 38.9 | 32.1 |
| UNO+ Fini et al. (2021) | DINOv1 | 68.6 | 98.3 | 53.8 | 70.3 | 95.0 | 57.9 | 35.1 | 49.0 | 28.1 |
| ORCA Cao et al. (2021) | DINOv1 | 81.8 | 86.2 | 79.6 | 73.5 | 92.6 | 63.9 | 36.3 | 43.8 | 32.6 |
| GCD Vaze et al. (2022) | DINOv1 | 91.5 | 97.9 | 88.2 | 74.1 | 89.8 | 66.3 | 51.3 | 56.6 | 48.7 |
| DCCL Pu et al. (2023) | DINOv1 | 96.3 | 96.5 | 96.9 | 80.5 | 90.5 | 76.2 | 63.5 | 60.8 | 64.9 |
| Prompt CAL Zhang et al. (2023) | DINOv1 | 97.9 | 96.6 | 98.5 | 83.1 | 92.7 | 78.3 | 62.9 | 64.4 | 62.1 |
| UNIC (ours) | DINOv1 | 94.04 | 94.24 | 93.94 | 84.55 | 93.39 | 80.11 | 44.35 | 51.63 | 40.70 |
| kMeans | DINOv2 | 93.32 | 83.84 | 96.55 | 77.64 | 84.28 | 74.31 | 69.22 | 70.91 | 68.36 |
| SimGCD Wen et al. (2023) | DINOv2 | 98.76 | 96.96 | 99.66 | 88.5 | **96.2** | 84.6 | 74.9 | 78.5 | 73.1 |
| SPTnet Wang et al. (2024) | DINOv2 | 97.80 | 98.12 | 97.48 | 90.1 | 96.1 | 87.1 | 76.3 | 79.5 | 74.6 |
| UNIC (ours) | DINOv2 | 98.02 | **97.95** | 98.05 | **90.86** | 95.32 | **88.63** | **81.36** | **84.52** | **79.78** |

match the GCD literature (Vaze et al., 2022). We utilize the labels of a fraction of images as per the GCD setting. See Appendix A.2 for details.

The proposed system is evaluated on the test splits of ImageNet and STL for clustering using the target labels. The evaluation for GCD is done with the target labels of the unlabelled portion of the training set. While the most early work in GCD has utilized the labelled test split of the dataset as an early stopping strategy (Wen et al., 2023), we exclude this to maintain a realistic open world setting.

In terms of hyperparameters, we report the results from $\tau_1 = 10$, $\eta = 70$ for ImageNet-100 (GCD), CIFAR-10, and STL-10. We report results obtained by $\tau_1 = 50$, $\eta = 1500$ for ImageNet splits (clustering). We conduct all experiments on a Nvidia RTX4000 GPU with 16GB of vRAM.

### 4.2 RESULTS

First, we provide results for clustering. While competitive with TEMI for STL-10, our method largely outperforms the prior work for the majority of metrics on the ImageNet splits. Notably, while the prior work does not outperform properly-tuned k-means consistently (see TEMI on ImageNet-50, for example), ours does in all cases. Also, since we finetune the last block of the backbone, we can even improve the k-means results. We investigate this phenomenon further in Section 4.4.

Not only do we achieve SOTA clustering results, our UNIC also performs favorably on GCD. Table 2 shows that while not the best method for CIFAR-10, we are still competitive with other works. Furthermore, for the more realistic (higher resolution images, more classes) dataset, we achieve SOTA performance on the unlabelled data ("All Classes"), primarily because of our methods strong performance on "Old Classes" (unlabelled images from seen classes). Our model's reliance on neighbor quality can sometimes hurt performance. However, in the case of GCD it is clearly a major benefit, since we have perfect neighbors for half of the images for "Old Classes." We examine our reliance on neighbor quality further in Section 4.4.

### 4.3 ABLATIONS

We conduct ablations to understand three items. Firstly, we validate our hypothesis about second-order neighborhood sizes being a proxy for the percentage of true positives in the first-order neighborhood. This analysis is present in Figure 4. It shows that we are able to pick a threshold for second-order neighborhood size (a point in x axis) such that we mostly get true positive neighbors (y value of the red line) for a large enough fraction of the dataset (y value of the blue line). More information and visualization for the phenomena is given in Appendix A.3.

Secondly, we examine the utility of the multiple loss terms proposed in our pipeline ($L_{pos}, L_{neg}, L_{ent}$). The results are given in Figure 5. The dotted red line above the dotted green line shows the gain from the proposed $L_{neg}$ term. More importantly, the other three lines show how $L_{neg}$ by itself is enough to converge to a stable solution. This is the first success in clustering literature to get rid of the $L_{ent}$ term. In addition, the trends show that we can further improve the performance

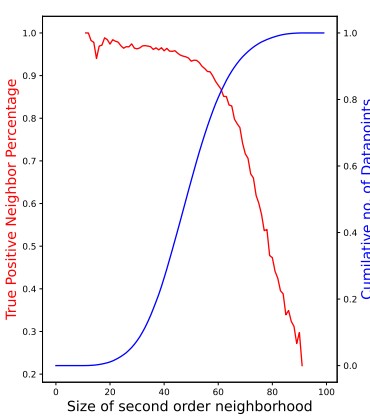

Figure 4: The variation of True positives with the size of the second order neighborhood

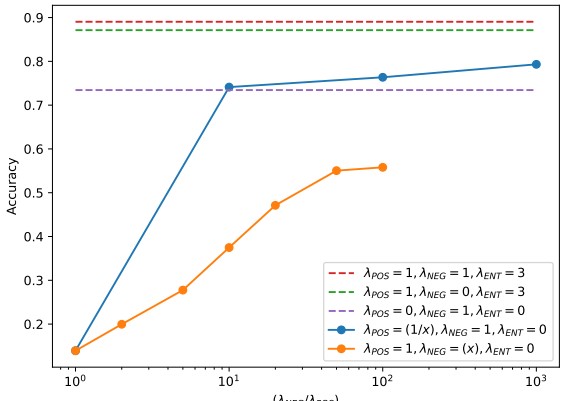

Figure 5: The variation of the accuracy of Imagenet-50 Clustering with different $\lambda$ values

Table 3: **Clustering Ablations** for ImageNet-50 with combinations of negative neighbor mining, entropy loss, and contrastive loss with $\lambda_{POS} = 1, \lambda_{NEG} = 1, \lambda_{ENT} = 3, \lambda_{CON} = 5$ after 50 epochs

| Positives | Negatives | Entropy | Contrastive | Accuracy |
|:---:|:---:|:---:|:---:|:---:|
| ✓ | | | | 2.03 |
| ✓ | ✓ | | | 17.64 |
| ✓ | | ✓ | | 89.48 |
| ✓ | ✓ | ✓ | | **90.12** |
| ✓ | ✓ | | ✓ | 14.12 |
| ✓ | ✓ | ✓ | ✓ | 88.24 |

Table 4: **GCD Ablations** (ImageNet-100) with various configurations of neighbors after 50 epochs. Supplementing the labeled positives and negatives with both *mined* (not random) negatives and *cleaned* positives helps performance.

| Positive Neighbors | | Negative Neighbors | | Accuracy | | |
|:---:|:---:|:---:|:---:|:---:|:---:|:---:|
| $D_L$ | $D_U$ | $D_L$ | $D_U$ | **All** | **Old** | **New** |
| Labeled | Mined | Random | Random | 82.22 | 92.36 | 77.13 |
| Labeled | Cleaned | Mined | Mined | **83.22** | 91.97 | **78.82** |
| Labeled | Cleaned | Labeled | Mined | 82.66 | **92.28** | 77.83 |

by the choice of $L_{neg}/L_{pos}$. This is further demonstrated with the results in Table 3. In addition, we also find that using a contrastive loss, as in some prior works, actually makes our clustering slightly worse. This can be attributed to the better supervision signal coming out of the proposed clustering head compared to the signal coming from the DINO contrastive head.

Thirdly, we conduct ablation studies to further analyze the effect of neighbors and their labelling on performance, we compare the impacts of random, mined, cleaned, and ground truth (labeled) neighbors for the two sets of images in GCD (labelled and unlabelled). In Table 4, we find that mining and cleaning positives for new classes improves upon the result we obtain by using the ground truth positives for old classes. In addition, we see improvement by mining negatives as per our algorithm, as opposed to using only ground truth neighbors or random neighbors. The random negative neighbors perform worse due to the false negatives in the picked sets as per Figure 8. Picking labeled negatives for $D_L$ causes an over-representation of negatives from old classes for these anchor images. This is further explained in Appendix A.13

### 4.4 ANALYSIS

We first show how our method converges, depending on the selection of certain key hyperparameters, in Figure 6. We show that we carefully choose our batch size, clustering head design, and level of finetuning to optimize performance. As mentioned previously, since we partially finetune the ViT backbone, its learned representation quality improves. We measure this improvement by tracking the k-means performance across training time in Figure 7. So, we conclude that while we target these classification tasks (GCD and clustering) directly, our method is indeed in some forms a useful strategy for unsupervised representation learning, at least as a finetuning method.

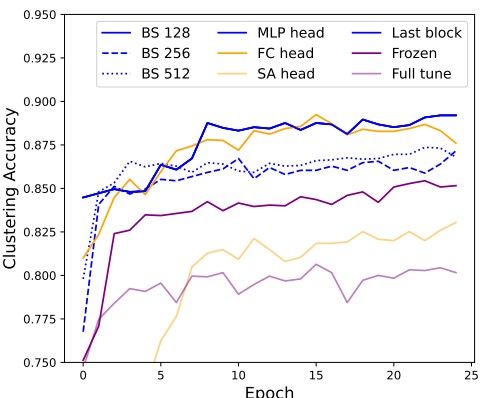
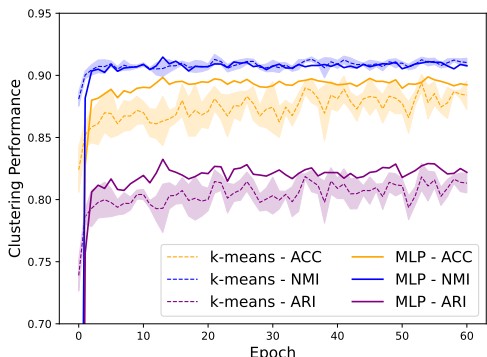

Figure 6: **Convergence Behavior** for clustering on ImageNet-50. We compare batch sizes, levels of finetuning (frozen, full-finetune, last block), and clustering heads (MLP, fully-connected, and self-attention, described in the appendix).

Figure 7: **k-Means Accuracy Over Time.** We plot the results of computing k-means for ImageNet-50 on the features extracted at different epochs during training. Notably, the learned representations form better clusters as we continue to train the backbone with UNIC.

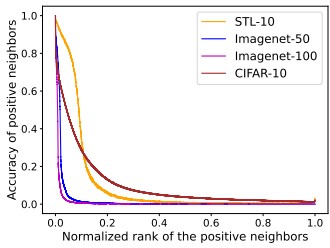
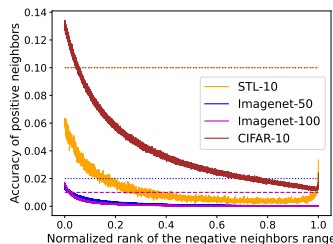
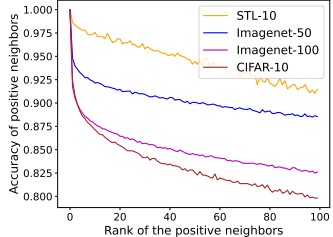

Figure 8: **Neighbor Analysis.** We show the amount of times a potential neighbor belongs to the same class as an anchor image ("Neighbor Accuracy"), depending on their relative distance, with the normalized index of the potential matching image in a sorted list of all images plotted on the x-axis (left). Notice there is a certain point where the amount of matches is comfortably low – we mine negative neighbors from this range. We zoom in on these negative neighbors, and compare the accuracy (which we want to be low) to random sampling, represented by the horizontal dashed lines (middle). We also zoom in on the positive neighbor range (right).

To select our threshold for negative neighbors, we consider the set of images. For each image, consider its distance from all other images, and sort them accordingly. For every image in the dataset, we thus have a sorted list of neighbor images, and each list is of the same length (containing all images in the dataset except the anchor). Now for each index we count how often the image belongs to the same class as the anchor, and plot this in Figure 8. This reveals how we mine our negative neighbors. Contrary to positive mining, we are not limited to some $k$ neighbors – instead, we can mine ~~around~~ more than half of all images as negatives, for any given anchor. Figure 8 (middle) validates our heuristical choices for $\tau_2$ because the false negatives are much lower than the dotted lines for any sufficient enough $\tau_2$. It should be noted that we do not tune $\tau_2$ extensively inorder to stay true for the problem settings. We show the reliability of the positive neighbors in the Figure 8 (right), giving some intuition as why we tend to set $10 \leq \tau_1 \leq 50$.

For cleaning, we show how we remove more neighbors as we increase the union sizes in Figure 9. Then, we notice that for smaller unions, our positive neighbors tend to be more "pure" in Figure 10. That is, they frequently belong to the same class, and it is more rare that positive neighbors belong to different classes. As expected, when we use these more pure positive neighbors, we get better clustering performance, which we show in Figure 11. This holds across different numbers of neighbors.

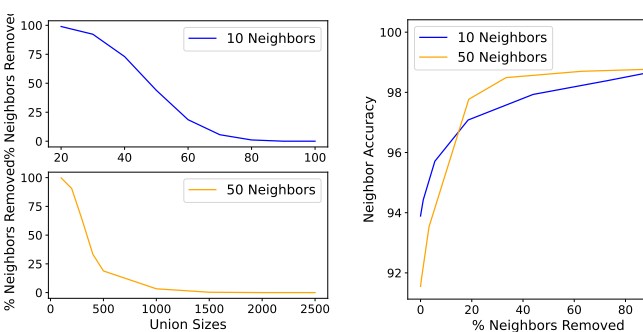 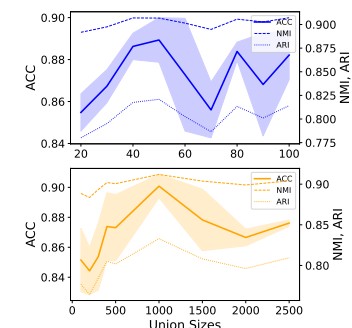

Figure 9: **Neighbors Removed by Union Size** for $\tau_1 = 10$ and $\tau_1 = 50$ on ImageNet-50.

Figure 10: **Neighbor Purity by Samples Removed** for $\tau_1 = 10$ and $\tau_1 = 50$ on ImageNet-50.

Figure 11: **Results by Union Size** for $\tau_1 = 10$ (top) and $\tau_1 = 50$ (bottom) on ImageNet-50.

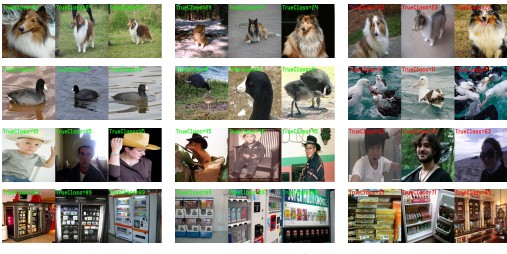 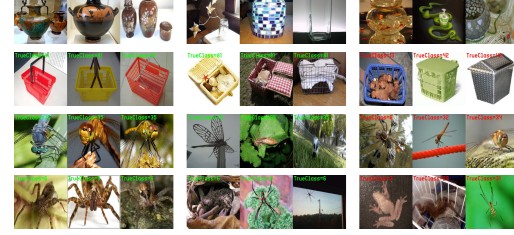

(a) Clustering Examples for ImageNet-100      (b) GCD Examples for ImageNet-100

Figure 12: **Cluster examples.** Each row corresponds to a cluster, for which we show triplets, from left to right, of the highest confidence true positives, the lowest confidence true positives, and randomly-sampled false positives. We find that in both cases UNIC is able to identify salient visual similarities, even when these are challenging, such as the lizard in the cowboy hat on the 3$^{\text{rd}}$ row (a).

We find that UNIC learns sensible clusters both for clustering and GCD in Figure 12. It tends to group things along the intended class boundaries, even when samples are strange, such as the instance with the lizard wearing the cowboy hat. Most false positives are visually similar to the true positives.

## 5 CONCLUSION

In this paper, we present an approach that improves on state-of-the-art for image clustering and GCD. Due to its flexibility and general formulation, our approach also conceptually unifies these two related vision problems. We also provide robust analysis of our major contributions, namely neighbor cleaning, negative neighbor mining, and other relevant hyperparameters. We hope our work here serves to break down the barriers between different levels of supervision, as we work towards real-world solutions that use labels in efficient, intuitive ways while still working well when labels are totally unavailable.

**Limitations:** Our proposed UNIC leverages neighbor information very well. However, if the neighbor mining fails, or is too noisy, our method fails. So, future work could address leveraging the neighbors in a manner that is less sensitive to noise.

**Future Work:** Our work has shown that deep clustering does not require an entropy maximization objective. Therefore, future work can relax the assumption on equally distributed classes and attempt to solve unbalanced clustering problems with UNIC (at $\lambda_{ENT} = 0$). Most ideas in computer vision start with image classification and then get extended into object detection. Similarly, UNIC's losses can be applied to open world object detection.

**Broader Impacts:** Image recognition can have negative impacts with respect to use in surveillance and autonomous weapons. Alternatively, good image clustering can help with medical imaging, where labels are scarce, for tracking efforts with endangered species, and other such applications.

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

# A  APPENDIX / SUPPLEMENTAL MATERIAL

## CONTENTS

## A.1  PREAMBLE

We give some details that we exclude from the main paper due to space constraints. We give information about the datasets (Appendix A.2) being used. Then we show a visualization of the hypothesis we use for the data distribution in the high dimensional space in Appendix A.3. Then we answer some frequently asked questions about the methodology being proposed with respect to novelty and fairness of comparison (Appendix A.4, A.6, A.8, A.5). We explain the empty spaces in clustering results in Appendix A.7. Then, we discuss the absence of contrastive loss in the SOTA model in Appendex A.9. We describe hyperparameters for UNIC and clustering head design for the alternative heads mentioned in Figure 6, in Appendix A.10. We also extend Figure 12 with more examples for GCD in Appendix A.11. Finally, we provide more examples for clustering in Appendix A.12.

## A.2  DATASET STATISTICS

Table 5: **Dataset Statistics** for our chosen tasks, Image Clustering and GCD.

| Dataset | # Classes | Image Clustering | | GCD | | |
| --- | --- | --- | --- | --- | --- | --- |
| | | Train Examples | Test examples | Old Classes | New Classes | Old labeled % |
| CIFAR-10 Krizhevsky et al. (2009) | 10 | - | - | 10 | 10 | 50% |
| STL-10 Coates et al. (2011) | 10 | 5k | 8k | - | - | - |
| ImageNet-50 Deng et al. (2009) | 50 | 64k | 2.5k | - | - | - |
| ImageNet-100 Deng et al. (2009) | 100 | 128k | 5k | 50 | 50 | 50% |
| ImageNet-200 Deng et al. (2009) | 200 | 256k | 10k | - | - | - |
| CUB-200 | 200 | - | - | 100 | 100 | 50% |

## A.3  VISUALIZATION OF SECOND ORDER NEIGHBORHOODS

The mechanism for cleaning neighbors based on second-order neighborhoods has been introduced in Figure 2 and Equation 4. In addition, this intuitive idea is experimentally shown in Figure 9 with

real data statistics. In this subsection, we try to visualize the phenomena within the Imagenet-100 dataset. The objective of this analysis is to remove the false negatives within first-order neighbors. The data points which are in the decision boundary of clusters will have this issue. However, these data points on the decision boundary should be identified without having access to the ground truth labels.

In order to demonstrate this, we pick two classes, embedd the data points onto the DINO space (which is 768 dimensional), and perform dimension reduction up to 2 dimensions (using PCA) for visualization. We try to show important information here.

- The first-order neighborhood of datapoints inside clusters will have high-quality (less noisy) positive neighbors.

- The second-order neighborhood size is a good proxy to figure out the data points at the decision boundaries.

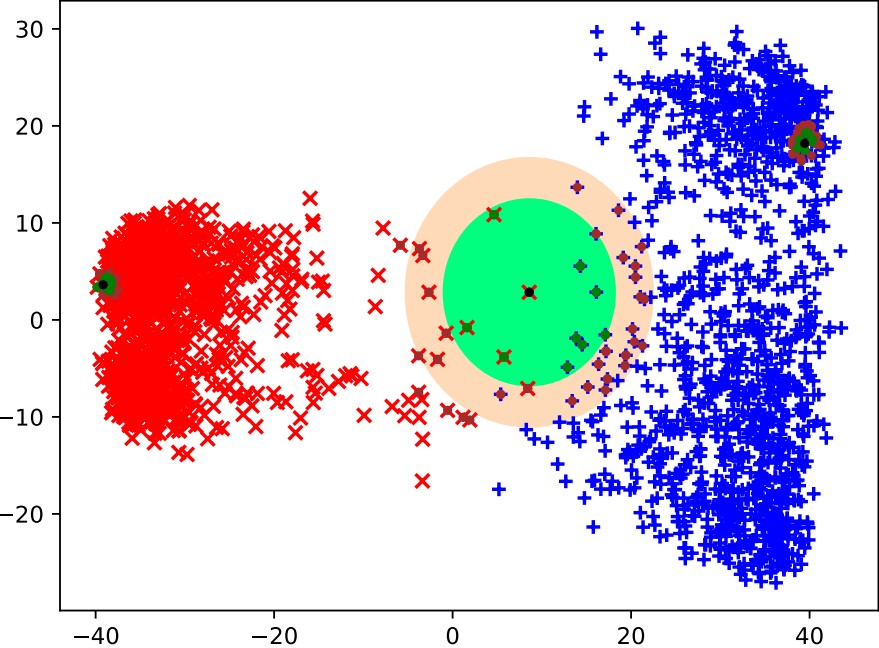

Figure 13: Neighbor cleaning based on second order information

Fig 13 shows the embedding space for two classes in Imagenet 100 dataset. The class labels are denoted by blue pluses ✚ and red crosses ✖ for the reader. Please note that they are **not** available for the algorithm. Consider three image embeddings denoted in black circles (left ✖, middle ✖ and right ✚). The first order neighbors are denoted in green circles ● and second order neighbors are denoted in brown circles ●. The second-order union sizes of left and right are small enough for them to be included. Therefore, the first order neighborhoods of left and right are included. The second-order union size of the middle ▉ is large so that the first order neighborhood of the middle ▉ is excluded. This cleaning startergy is effective because the green region of left contains red crosses only ✖ and the green region of right contains blue pluses ✚ only. The green region of middle contains both croses and pluses ✖✚ . Therefore, excluding this reduces noise in positive neighbors.

### A.4 FAQ(1) IS THIS BETTER THAN SCAN+DINO?

SCAN (Gansbeke et al., 2020) is a seminal work in Deep Clustering. We acknowledge that certain components of our work ($L_{POS}$, $L_{ENT}$) are inspired by SCAN.

SCAN was published in 2020 with a ResNet backbone. The absence of numbers in ImageNet columns for most ResNet based work in Table 1 shows that SCAN was one of the few work that was scaleable enough for larger datasets beyond 100 classes.

Few recent methods (Adaloglou et al., 2023; Zhou & Zhang, 2022a) have been proposed for image clustering using ViT-B/16 backbone. They have outperformed earlier ResNet work by a large margin.

Firstly, we show running kMeans on ViT-B/16 features is enough to outperform every work in the literature for Imagenet (subset) scale datasets. Then, we implement SCAN with ViT-B/16 and show it would be the state of the art in most clustering tasks even today.

However, the last row of Table 1 clearly shows that our work achieves better performance than SCAN+ViT-B/16.

In addition, we expand on the differences between SCAN and our work in Table 6. UNIC has a sequence of design decisions that improves the performance with respect to SCAN. Also, the utility of multiple heads in SCAN is not scalable when it comes to tuning the ViT backbone due to higher memory requirements.

Table 6: **UNIC Ablation on ImageNet-50**

| | Algorithm | ACC | NMI | ARI |
|---|---|---|---|---|
| $L_{SCAN}$ | +Resnet + heads = SCAN (Gansbeke et al., 2020) | 76.8 | 82.2 | 66.1 |
| $L_{SCAN}$ | +VitB/16 + heads | 85.48 | 88.58 | 78.19 |
| $L_{SCAN}$ | +VitB/16 | 83.33 | 88.65 | 76.81 |
| $L_{SCAN}$ | +VitB/16 + heads + Tune | Memory error | | |
| $L_{SCAN}$ | +VitB/16 + Tune | 87.12 | 90.18 | 80.45 |
| $L_{SCAN}$ | +VitB/16 + Tune + $L_{neg}$ | 89.04 | 90.70 | 82.19 |
| $L_{SCAN}$ | +VitB/16 + Tune + cleaning = UNIC | 90.80 | 91.81 | 84.25 |

### A.5 FAQ(2) WHAT IS THE NOVELTY OF THIS PAPER IF NEIGHBOR INFORMATION HAS ALREADY BEEN USED FOR CLUSTERING?

Almost every work in clustering literature has used positive neighbors mined by the Euclidean distance in embedding space. However, our work improves upon this idea on multiple fronts. Positive neighbor mining is cleaned by thresholding with respect to second-order neighborhood sizes. It should be noted that previous work (Gansbeke et al., 2020; Adaloglou et al., 2023) has run simulations with perfect positive neighbors (assuming they have an oracle to get this information) and showed that it would improve the clustering performance. However, UNIC is the first attempt at getting there. In addition, negative neighbor mining is also a novel contribution in utilizing neighbor informaiton.

### A.6 FAQ(3) IS IT POSSIBLE TO ATTRIBUTE THE PERFORMANCE GAINS TO NEW BACKBONES AND PERTAINING ONLY?

The naive performance of the backbone for both clustering and GCD can be seen with the kMeans result. The gain over the kMeans result is what a particular work contributes to. Interestingly, when the backbones are of moderate performance, there is much room for a proposed algorithm to contribute. This is seen by the larger gains among the ResNet based algorithms in Table 1. However, when the backbone has great performance, algorithms should be designed carefully to attain performance gains. Our results in Table 1 and Table 2 shows how UNIC has performance gains over kMeans in this challenging situation as well.

## A.7 FAQ(4) WHY ARE THERE MANY EMPTY CELLS IN THE CLUSTERING RESULTS TABLE?

Earlier deep clustering methods were not scale-able for a large number of classes. They usually attempted to solve CIFAR-10 like datasets or subsets of CIFAR-100 (20 classes out of the full 100), and Imagenet (10 classes out of the full 1000). Scaling them to present day benchmarks (up to 200 classes) requires significant engineering effort.

## A.8 FAIR COMPARISON WITH RESPECT TO BACKBONES

Unsupervised and semi-supervised (including GCD) work depends on a pre-trained backbone to bootstrap their frameworks. It is impossible to start with a randomly initialized backbone. The performance of an algorithm is highly dependent on the backbones being used.

Firstly, the fairness of comparison aspect of choosing backbones will be considered. Image clustering is an older problem that was first explored with ResNet backbones since 2020. The performance of the models made a significant jump once ViT backbones started being used since 2022 for the task. Inorder to provide a fair comparison, we re-implement important work from ResNet years (SCAN (Gansbeke et al., 2020)) with a ViT-B/16 backbone pretrained with DINOv1. There are no clustering papers that uses a DINOv2 backbone.

GCD was defined more recently in Vaze et al. (2022). Therefore, there is no ResNet-based work on the problem. Earlier work reported SOTA with ViT-B/16 DINOv1. More recent papers started reporting numbers for ViT-B/16 DINOv2 (with clear gains over DINOv1 numbers). We report our numbers with both DINOv1 and DINOv2 backbones. Our method outperforms previous methods with +1.45% accuracy on Imagenet-100 with DINOv1. Our method achieves SOTA over all (older and newer) methods with DINOv2 backbone at +0.75% on Imagenet-100 and +5.06% on CUB-200. It should be noted that DINOv2 CIFAR-10 numbers were not reported in the literature and we have re-run their code to generate those numbers.

Secondly, the backbone also dictates which type of frameworks (or components of frameworks) can be applied. For example, multiple heads and self-labelling like ideas from SCAN cannot be applied to ViT-B/16 DINO backbones. We have recreated SCAN for modern backbones by carefully changing these components of the original algorithm. Similarly, modern work like TEMI (Adaloglou et al., 2023) reports very low numbers for older ResNet MoCo backbones.

## A.9 CONTRASTIVE LOSS

Earlier work in the domain has used Supervised Contrastive loss (Vaze et al., 2022) to improve the performance. We conduct experiments with a similar setting and show it is redundant for the tasks being studied in this work. The results are given in Table 3.

For this ablation, we use the Supervised Contrastive loss coupled with the Dino-head projection of backbone features. We use the weighting term 5.0 to add this to the overall loss function and jointly train the full pipeline.

## A.10 HYPERPARAMETERS

Figure 6 analyzes ImageNet-50 clustering performance of the proposed system with respect to three hyperparameters – batch sizes, clustering head architectures, and freezing strategies. This subsection describes the experiments in detail.

Our base experiment runs on a batch size of 128 to be consistent with the rest of the GCD literature. The ablations results for batch sizes 256 and 512 are inferior to 128. However, it should be noted that the ImageNet-200 clustering result was obtained for the batch size of 512. Batches of smaller sizes are not representative of the full dataset statistics when it comes to a 200-class case. Specifically, this is detrimental to the optimization of the entropy-based loss function.

We experiment with three clustering heads. All these take Dino ViT-B/16 features (unnormalized) as input. The base experiments are run with the 2-layer perceptron (abbreviated MLP) with an intermediate dimension 2048. FC refers to the single, fully-connected linear layer. SA refers to a

self-attention-based classification architecture with patch size $16 \times 16$ and intermediate MLP dimension 2048.

In terms of training the backbone itself, we find it helps to partially finetune it (see Figure 6). We find that freezing the whole backbone (training none of it) is too restrictive. We find that fully-finetuning is also suboptimal. So instead, we train only the final (12th) transformer block.

### A.11 VISUAL EXAMPLES GCD

Some visual examples for GCD task on the ImageNet-100 dataset is given in this section. Figure 14 gives examples for old classes from GCD. Similarly, Figure 15 has examples from new classes.

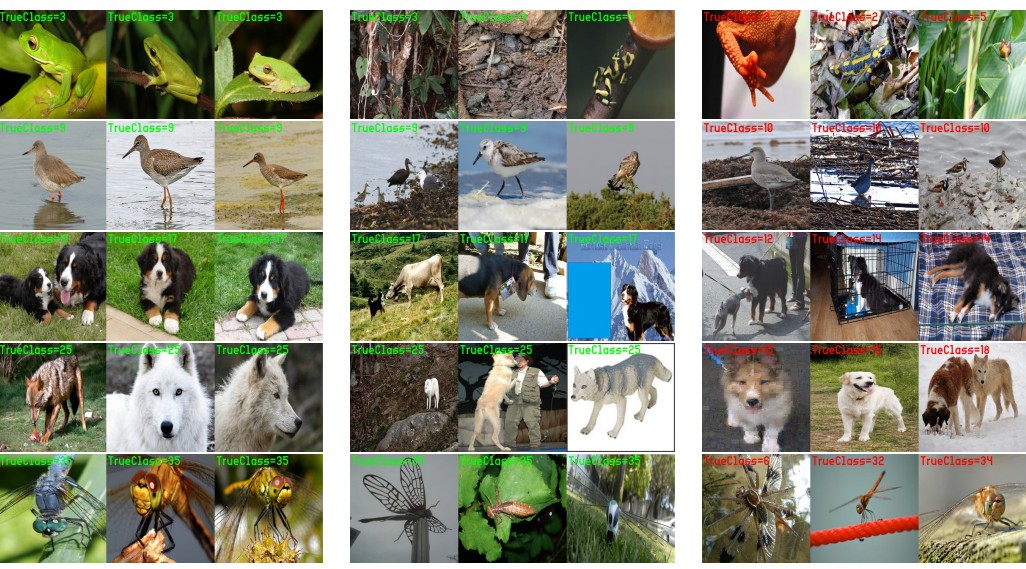

Figure 14: Example images from ImageNet-100 GCD for old classes. Left : highest confident true positives; Middle: least confident true positives; Right: False positives.

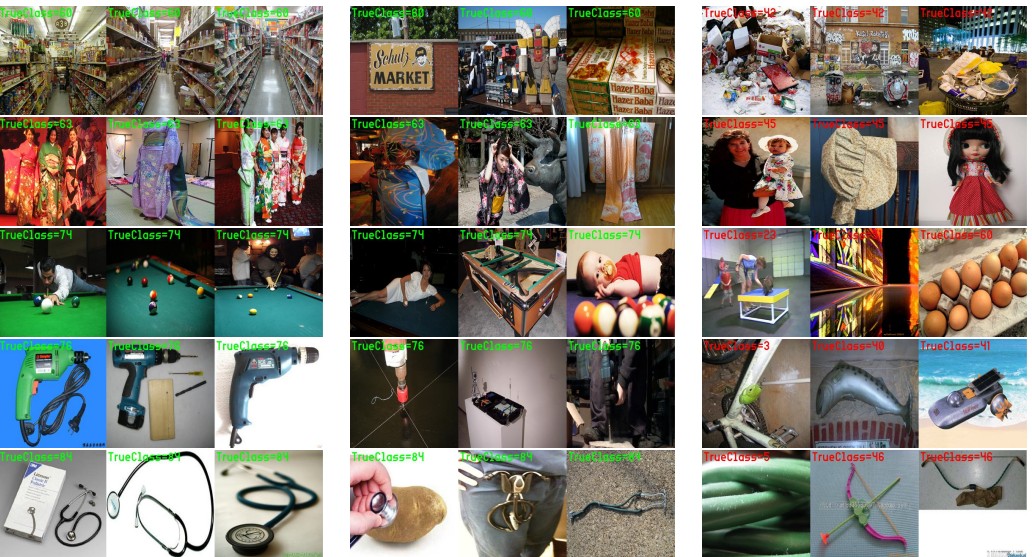

Figure 15: Example images from ImageNet-100 GCD for new classes. Left : highest confident true positives; Middle: least confident true positives; Right: False positives.

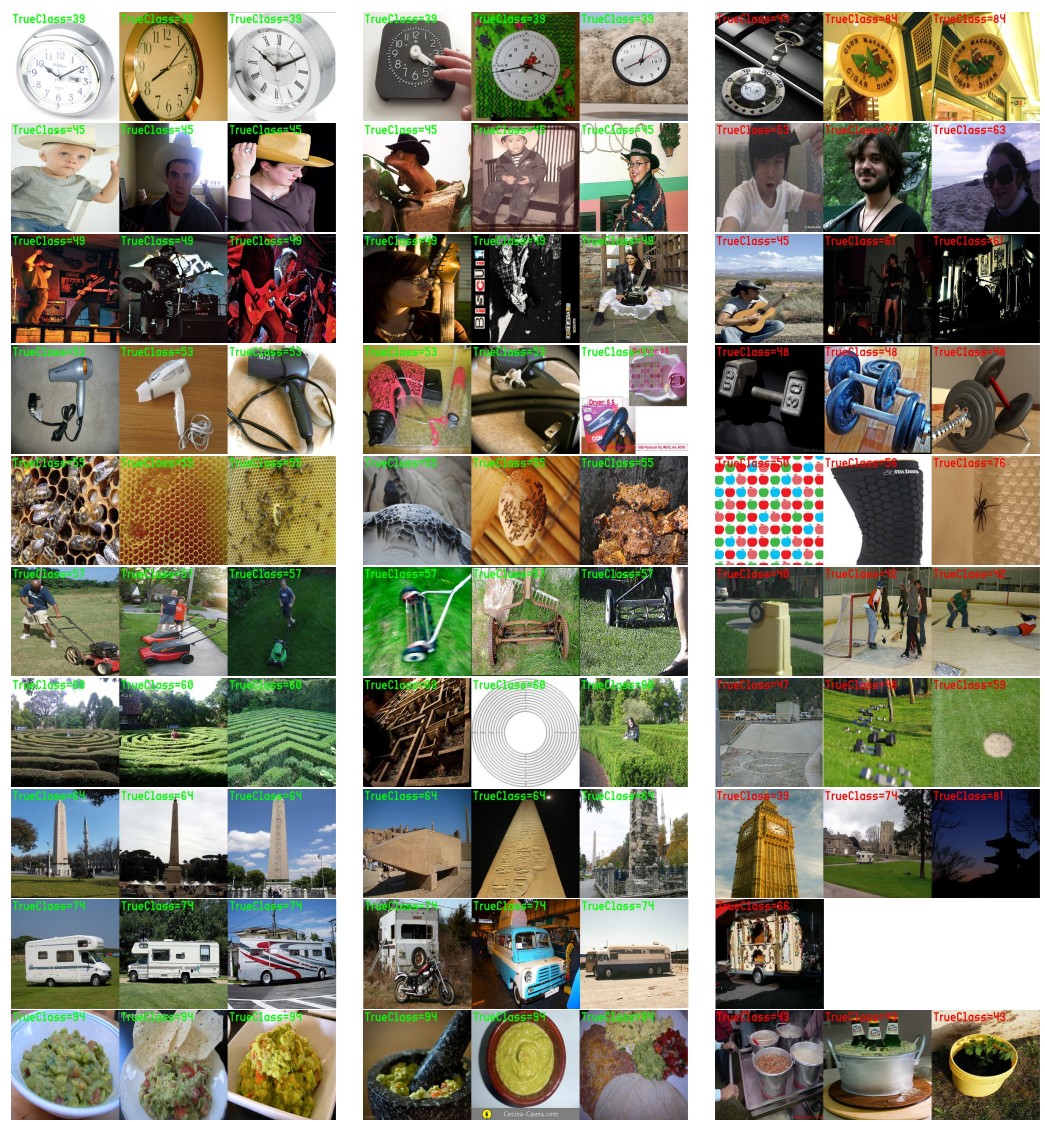

Figure 16: Example images from ImageNet-100 clustering. Left : highest confident true positives; Middle: least confident true positives; Right: False positives.

## A.12 VISUAL EXAMPLES – CLUSTERING

Figure 16 gives more examples for ImageNet-100 clusters learned in the fully-unsupervised setting.

## A.13 FAQ: WHY ARE MINED NEGATIVES BETTER THAN LABELED NEGATIVES IN TABLE 4?

Labeled negatives will always have a better true negative percentage than mined negatives. Therefore, it is counter-intuitive to witness a result where the All Class Accuracy using the mined negatives (83.22%) is better than the All Class Accuracy using the labeled negatives (82.66%).

Firstly, it should be noted that the Old Class Accuracy of 92.28% in the labeled case is above 91.97% in the mined case. This result matches the primary intuition about the lower number of false negatives in the labeled case.

The counterintuitive drop comes when the New Class Accuracy drops from 78.82% of the mined case to 77.83% in the labeled case. Consider a dataset $D$ with $N$ images. $0.25N$ will be labelled

old, $0.25N$ will be unlabelled old, and $0.5N$ will be unlabelled new. Consider the case where every image gets $k$ negative neighbors.

Given the label agnostic mining, the unlabelled $0.75N$ will get an equal proportion of negative neighbors from Old and New classes. This will amount to $0.375Nk$ from each. Consider that $\alpha$ fraction of negative neighbors for the labeled images come from the rest of the labeled images (all true negatives). However, this set will only have images from Old classes. Therefore, $(1 - \alpha)$ fraction should be mined. This will have equal fractions of Old and New classes. However, in total, $0.25Nk \left( \alpha + \frac{1-\alpha}{2} \right)$ of the negative neighbors will be from Old classes and $0.25Nk \left( \frac{1-\alpha}{2} \right)$ will be from New classes. For any choice of $\alpha$ (our reported result is for 0.5), the Old classes will be overrepresented as negative neighbors in the training. The under representation of the New class is the major contributing factor to the reported accuracy drop for New Classes, and therefore for the overall accuracy.

### A.14 FAQ: How Does UNIC's Neighbor Mining Compare to Hard Negative Mining

MoCHi Kalantidis et al. (2020) and HCL Robinson et al. (2021) has demonstrated the effectiveness of an unsupervised hard negative mining step on top of the vanilla SSL pretraining. These works shows impressive image classification results for the fully supervised case compared to the vanilla SSL pretraining. This appendix subsection attempts to explore (a) whether such techniques are useful for UNIC's problem settings, and (b) the reasons for it.

Both HCL (built on SimCLR Chen et al. (2020a)) and MoCHi (built on MoCoV2 Chen et al. (2020c)) attempt to find hard negatives to perform contrastive learning. Their objective is for the images in the dataset to uniformly be spread out through the embedding space. To this end, MoCHI specifically measures "uniformity" as defined in Wang & Isola (2020). HCL calls this "optimal embedding" and proves theoretical guarantees.

Such uniform spread of embeddings can be classified using MLPs trained with fully supervised learning. This is why HCL and MoCHi outperform the SimCLR and MoCo on classification tasks. It should be noted that spreading out the dataset uniformly is detrimental to the naturally occurring decision boundaries (i.e. cluster structure) that are used in unsupervised and semi-supervised learning settings. Our design choices in UNIC are to do the opposite – bring embeddings of the same class together while increasing the distance between the embeddings for different classes. In essence, this creates clusters (with natural decision boundaries) instead of a uniform spread.

We experiment with unsupervised kMeans clustering on HCL and MoCHi embeddings and append the results in Table 1. It should be noted that these numbers are below the kMeans result from SimCLR for STL-10 (which is reported in the first row from Gansbeke et al. (2020)) and kMeans result from MoCoV2 for all datasets. Then, we run the full UNIC training pipeline on the MoCHi and MoCoV2 backbones and report results in Table 7. MoCHi performs worse than vanilla MoCoV2 as the base representation for UNIC.

All these experiments provide strong evidence of why contrastive learning with hard negatives is less desirable for UNIC use cases.

### A.15 UNIC with other backbones

Table 7: **UNIC Clustering Performance with Different Backbones on ImageNet-50**

| Pretraining | Backbone | kMeans | | | UNIC | | |
|---|---|---|---|---|---|---|---|
| | | ACC | NMI | ARI | ACC | NMI | ARI |
| MOCHi | ResNet-50 | 61.88 | 73.44 | 44.92 | 61.80 | 72.81 | 48.80 |
| MoCOV2 | ResNet-50 | 63.04 | 75.75 | 47.00 | 64.28 | 76.64 | 53.42 |
| SwAV | ResNet-50 | 65.32 | 74.27 | 47.07 | 73.20 | 80.74 | 61.86 |
| DINOv1 | ResNet-50 | 70.68 | 77.82 | 52.17 | 75.44 | 81.62 | 64.77 |
| DINOv1 | VitB/16 | 82.36 | 87.91 | 73.89 | 90.80 | 91.81 | 84.25 |
| iBot | VitB/16 | 82.48 | 86.94 | 68.10 | 85.04 | 89.45 | 77.70 |
| DINOv2 | VitB/14 | 87.20 | 89.22 | 60.42 | 94.60 | 95.09 | 90.77 |
| Supervised | VitB/16 | 89.76 | 90.12 | 66.08 | 96.44 | 95.99 | 93.18 |

The main text of the paper experiments with DINOv1 and DINOv2 backbones for UNIC and compare with the current state of the art. This allows us to evaluate the proposed solution on it's merits rather than getting an unfair advantage from the backbone's pretraining. The fairness aspect of this is explained in Appendix A.8. In this subsection, we run the UNIC pipeline with different backbones and report results in Table 7.

The results from Table 7 shows how UNIC can be used on top of every SSL-trained backbone we have experimented on. UNIC reports consistent gain over the kMeans result from the vanilla backbone for most cases. However, the gains are minimal when the vanilla backbone's performance is $< 65\%$. UNIC has significant gains for all better backbones. It should be noted that ViT backbones are generally better than ResNet backbones. Also, the best performance of UNIC is obtained when the backbone is supervised-trained.

These experimental results are strong evidence for UNIC's generalizability over a spectrum of weak backbones to very strong backbones.

