# OpenReview forum: "Utilization of Neighbor Information for Image Classification with Different Levels of Supervision"
_ICLR.cc/2025/Conference — Submitted to ICLR 2025_

### Official Review · Reviewer_zhtP · 2024-11-02

**Soundness:** 2
**Presentation:** 1
**Contribution:** 2
**Rating:** 3
**Confidence:** 5

**Summary:**

The paper presents **UNIC** (Utilizing Neighbor Information for Classification), a method designed to improve performance in both image clustering (unsupervised learning) and generalized category discovery (GCD) (semi-supervised learning setting). The key contributions and insights from the paper include: 1) Unified Framework for Clustering and GCD: UNIC bridges the gap between clustering and GCD by utilizing a shared pipeline, where it leverages nearest-neighbor relationships in the feature space. For clustering tasks, UNIC identifies and refines positive (same class) and negative (different class) neighbors to improve cluster quality. In the GCD setting, available labels serve as perfect neighbors for known classes, enhancing the model's performance on labeled and unlabeled data alike. 2) Innovative Neighbor Mining and Cleaning Strategy: The method introduces a neighbor-cleaning mechanism, which refines positive neighbors by filtering based on the union size of second-order neighbors, thus ensuring higher purity in the selected neighbor sets. 3) End-to-End Learning Pipeline: Unlike previous multi-stage clustering approaches, UNIC uses an end-to-end training pipeline that enhances representation learning without self-labeling steps, which can be inefficient.

The approach involves two key stages: 1) Neighbor Mining, which identifies positive and negative neighbors by proximity in the feature space. 2) Model Training, which fine-tunes a backbone model (ViT-B/16 pretrained with DINO) using classification and entropy-based regularization losses. Positive neighbors are pulled together, and negative ones are pushed apart to encourage meaningful clustering. UNIC demonstrates state-of-the-art performance across multiple datasets. For the clustering task, it achieves superior accuracy and normalized mutual information (NMI) on benchmarks like STL-10 and ImageNet-100. For the GCD task, it outperforms competing methods on datasets like ImageNet-100 and CUB-200, particularly excelling in settings with high-resolution images and diverse classes.

**Strengths:**

It's intriguing to see how this paper unifies image clustering and generalized category discovery within a single pipeline. Building on this concept, the authors propose a clustering approach that leverages the mining of positive and negative neighbors. This unified framework enhances model performance across various downstream tasks.

The paper includes extensive evaluations on several benchmarks, demonstrating clear performance improvements. Additionally, the ablation studies offer valuable insights into the contributions of each component in the framework.

End-to-End Learning Pipeline: Unlike previous multi-stage clustering approaches, UNIC uses an end-to-end training pipeline that enhances representation learning without self-labeling steps, which can be inefficient.

**Weaknesses:**

I have several concerns regarding the technical contributions of this paper. While I agree that positive and negative neighbor mining can improve clustering methods, this approach has already been extensively explored in previous contrastive learning research, such as in [1 ICLR2021] and [2 NeurIPS 2020].

The selection methods for positive and negative samples also appear somewhat simplistic. Using the nearest samples as positives and the furthest as negatives might not be a robust strategy. For instance, previous works, including [1], have shown that employing “hard” negative samples (those that are more challenging to distinguish from positives) can significantly improve model performance. Given the existing research on positive and negative sample mining, there are likely many unexplored approaches that could yield interesting results. It would be beneficial to explore how the selection of positive and negative samples could be refined for this task setting to improve efficiency and effectiveness.

Additionally, some key baselines, such as DeepDPM [3], are missing, and testing on larger datasets, like ImageNet-1k, would strengthen the evidence for model improvements. Many of the baselines included in this paper are not the latest or most competitive, which further limits the impact of the comparisons presented.

The paper’s presentation could also benefit from some adjustments. Placing additional figures, particularly those referenced in the main text, after the summary section disrupts readability.

Consistency in formatting could be improved as well; for instance, Table 2 contains results with varying decimal places, with some values displayed to two decimals and others to only one.

[1] Robinson, Joshua, Ching-Yao Chuang, Suvrit Sra, and Stefanie Jegelka. "CONTRASTIVE LEARNING WITH HARD NEGATIVE SAMPLES." In International Conference on Learning Representations (ICLR). 2021.

[2] Kalantidis, Yannis, Mert Bulent Sariyildiz, Noe Pion, Philippe Weinzaepfel, and Diane Larlus. "Hard negative mixing for contrastive learning." Advances in neural information processing systems 33 (2020): 21798-21809.

[3] Ronen, Meitar, Shahaf E. Finder, and Oren Freifeld. "Deepdpm: Deep clustering with an unknown number of clusters." In Proceedings of the IEEE/CVF Conference on Computer Vision and Pattern Recognition, pp. 9861-9870. 2022.

**Questions:**

I am curious if authors have tried various representation learning methods (such as MAE, Supervised ViT, iBot, etc) beyond these ones presented in the paper? What kind of self-supervised learning strategy can achieve the best performance? Do you have any insights on it?

---

> ### Author Response · Authors · 2024-11-19
> **Author response to Reviewer zhtP**
>
> We thank the anonymous reviewer for their time and effort in producing a meaningful review, a very detailed summarization of our work, and a comprehensive note of our strengths. Firstly, we will respond to the reviews along with an updated PDF for the questions that do not require further experimentation. Next, we have a set of clarification questions about the additional experiments/comparisons requested by the reviewer.

---

> ### Author Response · Authors · 2024-11-19
> **[W 4.3.1]**
>
> [W 4.3.1] Additionally, some key baselines, such as DeepDPM [3], are missing,
> --
> This baseline is present in Table 01 (4th row of results).

---

> ### Author Response · Authors · 2024-11-19
> **[W 4.3.3]**
>
> [W 4.3.3] Many of the baselines included in this paper are not the latest or most competitive, which further limits the impact of the comparisons presented.
> --
> While we acknowledge the possibility of missing something, we have tried to add (1) key papers from the earlier literature and (2) almost all the competitive papers from recent literature for comparison purposes. We would highly appreciate it if the anonymous reviewer could point us to the literature that we have missed for comparison purposes.

---

> ### Author Response · Authors · 2024-11-19
> **[W 4.6]**
>
> [W 4.6] Consistency in formatting could be improved as well; for instance, Table 2 contains results with varying decimal places, with some values displayed to two decimals and others to only one.
> --
> When re-using numbers from other works, we are limited by the precision they use. We provide 2 decimal precision whenever available. The 2nd decimal place is required to assess the difference between ViT-B/16 results of TEMI, SCAN, and kMeans.

---

> ### Author Response · Authors · 2024-11-21
> **[W 4.4]**
>
> [W 4.4] The paper’s presentation could also benefit from some adjustments. Placing additional figures, particularly those referenced in the main text, after the summary section disrupts readability.
> --
> We request some clarification as to whether this comment is about referring to figures from the appendix in the main text. If not, we would be highly appreciative of specific suggestions on how to improve our presentation and readability.

---

> ### Author Response · Authors · 2024-11-28
> **[W 4.1]**
>
> [W 4.1] I have several concerns regarding the technical contributions of this paper. While I agree that positive and negative neighbor mining can improve clustering methods, this approach has already been extensively explored in previous contrastive learning research, such as in [1 ICLR2021] and [2 NeurIPS 2020].
> --
> We thank the reviewers for bringing up this question. The existing work in the hard negative literature shows gains for a fully supervised image classification setting. However, these techniques fail to improve UNIC’s problem settings.
>
> We append a MoCHi  [2 NeurIPS 2020] row to Table 1 and Table 7. We also append an HCL [1 ICLR2021]  row to Table 1. These results show the failure of hard negative mining to improve performance.
>
> Then we add an “Appendix A.14 FAQ: HOW DOES UNIC’S NEIGHBOR MINING COMPARE TO HARD NEGATIVE MINING?” to explain the reasons for this failure, why hard negative mining in literature is orthogonal to our work, and why we specifically need UNIC’s neighbor mining techniques.

---

> ### Author Response · Authors · 2024-11-28
> **[W 4.2]**
>
> [W 4.2] The selection methods for positive and negative samples also appear somewhat simplistic. Using the nearest samples as positives and the furthest as negatives might not be a robust strategy. For instance, previous works, including [1], have shown that employing “hard” negative samples (those that are more challenging to distinguish from positives) can significantly improve model performance. Given the existing research on positive and negative sample mining, there are likely many unexplored approaches that could yield interesting results. It would be beneficial to explore how the selection of positive and negative samples could be refined for this task setting to improve efficiency and effectiveness.
> --
>
> Please refer to our response to [W4.1]. We have given an explanation and experimental evidence as to why hard negative samples can improve performance in supervised settings (which is explored in the existing literature) but not in unsupervised or semi-supervised settings (which is explored in this paper). The UNIC’s task setting does not benefit from the existing literature on hard negatives for contrastive learning.

---

> ### Author Response · Authors · 2024-11-28
> **[W 4.6]**
>
> [W 4.6] I am curious if authors have tried various representation learning methods (such as MAE, Supervised ViT, iBot, etc) beyond these ones presented in the paper? What kind of self-supervised learning strategy can achieve the best performance? Do you have any insights on it?
> --
> We are thankful for this insightful question from the anonymous reviewer. We limited our main text to DINO representation learning methods to stay consistent with the literature. In order to answer this question, we ran additional experiments and reported the results in Table 7: UNIC Clustering Performance with Different Backbones on ImageNet-50.
>
>
> Here, we present results from both ResNet and ViT backbones trained on a range of SSL techniques. In addition, we report a result from a supervised trained backbone as a best-case scenario.
>
> The insights from these experiments are discussed in Appendix A.15 UNIC WITH OTHER BACKBONES.

---

> > ### Comment · Reviewer_zhtP · 2024-11-29
> >
> > Hi authors, thank you for your detailed responses. Most of my questions have been addressed, and I appreciate the effort in clarifying your approach. However, I still have concerns on the technical contributions of this paper, especially considering that many papers have explored the hard negative mining sampling. After the rebuttal, although I agree that there are some differences, the difference may not be sufficient enough for ICLR papers. Unfortunately, I may decide to keep my score.

---

> > > ### Author Response · Authors · 2024-12-03
> > > **Differences from Other Papers**
> > >
> > > We would like to kindly request other references. From those provided, our neighbor mining follows a different principle (mining reliable negatives, specifically avoiding the hard negatives) and thus is a different technique entirely, for a different task (clustering/GCD as opposed to instance discrimination), with a different objective. There is no overlap between our findings with [1] or [2].
> > >
> > > Considering we have addressed most questions, considering the strengths mentioned in the original review, and considering that our paper does not actually overlap these prior works, we kindly request the reviewer to reconsider the rating or to further clarify how our work, which proposes a framework and techniques to unify GCD and clustering, overlaps with works that do not provide methods that are competitive on either task.

---

### Official Review · Reviewer_HhpM · 2024-11-03

**Soundness:** 2
**Presentation:** 2
**Contribution:** 1
**Rating:** 3
**Confidence:** 5

**Summary:**

This paper approaches image clustering and Generalized Category Discovery (GCD) with a proposed union framework, named Utilizes Neighbor Information for Classification (UNIC).
A novel neighbor mining strategy is introduced to clean noise data points among the set of positive neighbors, and a general pipeline that can be trained end-to-end is designed as well.

**Strengths:**

S1. They design a simple and effective strategy to exclude noise data points in the set of positive neighbors, which utilizes the information of second-order Euclidean distance.
S2. The proposed method addresses image clustering and GCD simultaneously.
S3. The authors demonstrate the effectiveness of UNIC in their framework with sufficient experimental analysis. Their in-depth analysis shows that their proposed method clearly contributes to performance improvement.

**Weaknesses:**

W1. When encountered with fine-grained classes, proposed methods show significant reduction with a less powerful backbone, e.g., DINOv1. Could the authors analyze the bounds on the backbone’s feature extraction capabilities that make the proposed approach fail? For example, when the basic K-means clustering accuracy drops to 60% or 50%, the proposed UNIC will not work?

W2. For a comprehensive clustering ablation, could the author replace the proposed positive mining strategy with a simplified one or existing one and show it in Tab. 3 to validate the effectiveness of the proposed positive mining strategy?

W3. There should be more ablations on GCD in Tab. 4, such as “Labeled, Mined for D_L and D_U of positive neighbors, and Labeled-Mined for D_L and D_U of negative neighbors”

W4. Could the author explain the selection of τ_2 for ImageNet and STL? For CUB, what is the parameter τ_2 set to?

W5. For fine-grained image benchmarks, the experiments are only conducted based on the CUB dataset, which cannot demonstrate the generation ability of the model. The author should conduct experiments on other fine-grained like Stanford Cars and FGVC-Aircraft. If the proposed UNIC framework can well-mind clean neighbors, it should also achieve comparable performance with state-of-the-art benchmarks as well.

W6. There are several typos, such as Line 258 “y ̂_i,y ̂_p,y ̂_p”

**Questions:**

Please justify the issues in Weaknesses.

---

> ### Author Response · Authors · 2024-11-19
> **Author response to Reviewer HhpM**
>
> We thank the anonymous reviewer for their time and effort in producing a meaningful review. We will respond to the reviews along with an updated PDF. Firstly, we will respond to the questions that do not require further experiments. In the meantime, we will run additional experiments to answer the rest of the questions.
>
> While the summary and strengths have pointed out most of our contributions, we would like to emphasize that our strategy of mining negatives, and their analysis of their contribution to improving the accuracy of clustering and GCD tasks (or any tasks) is novel as well.

---

> ### Author Response · Authors · 2024-11-19
> **W2**
>
> W2. For a comprehensive clustering ablation, could the author replace the proposed positive mining strategy with a simplified one or existing one and show it in Tab. 3 to validate the effectiveness of the proposed positive mining strategy?
> --
> The authors have reported results from simpler mining strategies in Table 6 in the Appendix. The bottom row (7th row) of the table (90.80%) is obtained by our full mining strategy (which includes positive, cleaning, and negatives). The 6th row (89.04%) reports a simple strategy (which includes positives and negatives only). The 5th row (87.12%) reports the simplest strategy, which is to mine positives only.
>
> The authors would like the input of the anonymous reviewer as to whether this ablation should be present here as a separate table or be added to the main Table 6 on the paper.

---

> ### Author Response · Authors · 2024-11-19
> **W4**
>
> W4. Could the author explain the selection of τ_2 for ImageNet and STL? For CUB, what is the parameter τ_2 set to?
> --
> We have updated line 292 to give τ_2 values for all datasets. We append lines 472 to 475 to validate our choices for τ_2 in retrospect.
> The authors have not tuned τ_2 extensively in this study. We only pick a τ_2 that is intuitively big enough to cover the necessary region of the neighborhood based on the number of classes and example images per class in the dataset.

---

> ### Author Response · Authors · 2024-11-19
> **W6**
>
> W6. There are several typos, such as Line 258 “y ̂_i,y ̂_p,y ̂_p”
> --
> Thank you for pointing this out. We have corrected this.

---

> > ### Comment · Reviewer_HhpM · 2024-11-27
> > **Acknowledgement and wait for respones about other Questions(W1,W3,W5)**
> >
> > Thanks for the authors' efforts. These responses partially address my concerns. However, my main concerns about the proposed method's poor performance with the relatively weak backbone are still unsolved due to a lack of justification and experimental results. This limits the proposed method's generalization ability. Thus, I keep my initial rate.

---

> ### Author Response · Authors · 2024-11-28
> **W1**
>
> W1. When encountered with fine-grained classes, proposed methods show significant reduction with a less powerful backbone, e.g., DINOv1. Could the authors analyze the bounds on the backbone’s feature extraction capabilities that make the proposed approach fail? For example, when the basic K-means clustering accuracy drops to 60% or 50%, the proposed UNIC will not work?
> –
> We thank the reviewer for this insightful question. We conduct a range of experiments with different backbones to answer this question and report both kMeans and UNIC results for them in Table 7: UNIC Clustering Performance with Different Backbones on ImageNet-50 and explain the results in Appendix A.15 UNIC WITH OTHER BACKBONES.
> UNIC’s gains over kMeans drop to very low numbers (<1%, or negative) when the feature extraction is underperforming (MoCHi, MoCoV2) where kMeans accuracy itself is low 60%s. UNIC consistently performs well for representations for which kMeans accuracy is over 65%.

---

> > ### Comment · Reviewer_HhpM · 2024-11-29
> > **Discussion**
> >
> > Thanks for the author's detailed response. I'd like to explain my question further. The relatively low accuracy of UNICs on CUB is convincing for me, based on the more experimental results. However, I hope the authors can make an in-depth analysis to explore whether the UNIC cannot effectively improve CUB performance with underperforming features, which is a limitation of UNIC. Based on my understanding, reducing noise neighbors is the main idea of this paper. Intuitively, a better feature/backbone results in less noise compared with using underperforming features. However, the proposed method seems more effective for less-noise cases, which is strange to me. I am curious whether this observation happened in another fine-grained dataset in the GCD benchmark. I guess that the experimental results of W5 can provide more insights into this question.

---

> ### Author Response · Authors · 2024-12-03
> **W5**
>
> W5. For fine-grained image benchmarks, the experiments are only conducted based on the CUB dataset, which cannot demonstrate the generation ability of the model. The author should conduct experiments on other fine-grained like Stanford Cars and FGVC-Aircraft. If the proposed UNIC framework can well-mind clean neighbors, it should also achieve comparable performance with state-of-the-art benchmarks as well.
> --
>
> The authors thank the reviewer for this question. We acknowledge that the only FGCV result on the main text was where UNIC outperformed the literature only for the DINOv2 backbone case. We have conducted additional experiments during the rebuttal period to demonstrate how and when UNIC’s GCD capabilities are generalizable across the SSB suite.
>
>
> We run experiments on CUB-200, StanfordCars-196, and Aircrafts-100 datasets. We run two experiments for each dataset. First, we experiment with the DINOv1 pre trained Vit-B/16 backbone similar to earlier GCD literature. Then, we follow experiments with SimGCD [ICCV 2023] pretrarined backbone similar to recent GCD literature SPTnet [ICLR 2024]. Please refer to the table below for experimental results.
>
>
> Our results show that UNIC suffers from the low-quality neighbors mined with DINOv1 backbone. However, UNIC outperforms the existing literature when it is able to utilize SimGCD pretrained backbones and achieve state-of-the-art.
>
>
> It should be noted that SPTnet achieves its improvement over the previous literature by adding additional prompt parameters to the ViT backbone. UNIC achieves better performance without changing the backbone architecture (Only tuning the last block).
>
> | **Algorithm**       | **Backbone**       | **CUB-200** (All/Old/New) | **Aircrafts-100** (All/Old/New) | **SCars-196** (All/Old/New) |
> |----------------------|--------------------|----------------------------|----------------------------------|-----------------------------|
> | kMeans (Vaze et al., 2022) | DINOv1          | 34.3 / 38.9 / 32.1         | 12.9 / 12.9 / 12.8              | 12.8 / 10.6 / 13.8          |
> | UNO+ (Fini et al., 2021)   | DINOv1          | 35.1 / 49.0 / 28.1         | 28.3 / 53.7 / 14.7              | 35.5 / 70.5 / 18.6          |
> | ORCA (Cao et al., 2021)    | DINOv1          | 36.3 / 43.8 / 32.6         | 31.6 / 32.0 / 31.4              | 31.9 / 42.2 / 26.9          |
> | GCD (Vaze et al., 2022)    | DINOv1          | 51.3 / 56.6 / 48.7         | 45.0 / 41.1 / 46.9              | 39.0 / 57.6 / 29.9          |
> | DCCL (Pu et al., 2023)     | DINOv1          | 63.5 / 60.8 / 64.9         | -- / -- / --                    | 43.1 / 55.7 / 36.2          |
> | Prompt CAL (Zhang et al., 2023) | DINOv1      | 62.9 / 64.4 / 62.1         | 52.2 / 52.2 / 52.3              | 50.2 / 70.1 / 40.6          |
> | UNIC                  | DINOv1          | 44.6 / 51.6 / 40.7      | 25.5 / 28.6 / 23.9           | -- / -- / --                |
> | SimGCD (Wen et al., 2023)   | DINOv1          | 60.3 / 65.6 / 57.7         | 54.2 / 59.1 / 51.8              | 53.8 / 71.9 / 45.0          |
> | | | | | |
> | **SPTnet (Wang et al., 2024)** | DINOv1+SimGCD | 65.8 / 68.8 / **65.1** | 59.3 / 61.8 / **58.1**   | 59.0 / 79.2 / **49.3** |
> | **UNIC (ours)**       | DINOv1+SimGCD    | **69.4** / **83.4** / 62.4 | **63.9** / **68.1** / 61.8 | **62.6** / **90.7** / 49.0 |

---

### Official Review · Reviewer_JSTd · 2024-11-05

**Soundness:** 2
**Presentation:** 3
**Contribution:** 2
**Rating:** 3
**Confidence:** 4

**Summary:**

This paper proposes a novel neighbor mining strategy (UNIC) for both image clustering and generalized category discovery (GCD). The main idea is to firstly mine nearest neighbors as positive and negative ones, and then refine the positive neighbors based on second-order neighborhoods. Experiments show the proposed method has achieved promising performance on clustering and GCD settings.

**Strengths:**

The motivation for bridging clustering and GCD is interesting and reasonable.
The paper is well written and it is easy to follow the proposed method.
Competitive results on several benchmarks are achieved by the proposed method.

**Weaknesses:**

The main contribution in this work lies in how to mine positive and negative neighbors. However, the implementation details in the proposed UNIC are not innovative a lot. There have many related methods about improving the positive and negative candidates
for both clustering and GCD field. It is unclear about the main difference and contribution proposed in UNIC. Also, it is unclear why UNIC
can achieve significant improvements over prior works.

**Questions:**

It is encouraged to clarify the novelty in the proposed UNIC, as its implementation looks very common and similar to some existing works.

The experiments are not comprehensive. It is needed to compare with other positive/negative mining strategies directly.

Moreover, the experiments lack insightful analysis. For example, in Table 3, why the contrastive loss harms the performance of clustering;
For the GCD ablations in Table 4, it is curious why the labeled negative neighbors are inferior to the mined ones (when comparing the results in the second and third rows).

---

> ### Author Response · Authors · 2024-11-19
> **Author response to Reviewer JSTd**
>
> We thank the anonymous reviewer for their time and effort in producing a meaningful review. We will respond to the reviews along with an updated PDF. In addition, we have a set of clarification questions about the additional experiments/comparisons requested by the reviewer.

---

> > ### Comment · Reviewer_JSTd · 2024-11-27
> >
> > Thanks for the rebuttal from the authors. However, I am still concerned about the novelty and experiment results in the paper. Also, I agree with other reviewer who suggested comparing this work with prior works as well

---

> > > ### Author Response · Authors · 2024-12-03
> > > **Novelty and Experiments**
> > >
> > > Without clarification, it is difficult to take action on this feedback. In terms of novelty, what is not compelling from the clarification provided in this thread?
> > >
> > > In terms of comprehensiveness of experiments and explaining the results, these have been changed and added to as well. What, specifically, is lacking?
> > >
> > > We have compared with many prior works (to our knowledge, all relevant prior works), and added all relevant works suggested by reviewers. The negative mining works are the only works specifically mentioned that we do not compare to, numerically, and that is because they are general contrastive pre-training strategies, not methods for our tasks (GCD and clustering), nor can they be adapted as such.
> > >
> > > Considering how we have addressed all clearly communicated weaknesses, and the SOTA results (including those we conveyed in the thread with reviewer 48qz) across multiple tasks that previously had only been solved by specialized methods, we kindly request the reviewer to reconsider the rating.

---

> ### Author Response · Authors · 2024-11-19
> **[W 2.1]**
>
> [W 2.1] The main contribution in this work lies in how to mine positive and negative neighbors. However, the implementation details in the proposed UNIC are not innovative a lot. There have many related methods about improving the positive and negative candidates for both clustering and GCD field. It is unclear about the main difference and contribution proposed in UNIC. Also, it is unclear why UNIC can achieve significant improvements over prior works.
> --
> According to the best of our knowledge of the literature, there isn’t much work that explores improving positives for this style of clustering and GCD. Utilizing negatives (instead of randomly sampling from the whole dataset) has not been explored in any prominent work. We would be extremely grateful if the anonymous reviewer could point out such literature, which we could incorporate into our related work sections.
>
> Our reasoning for why UNIC achieved significant improvements in our novel contributions – (1) positive cleaning strategy, (2) the use of negatives, and our improvements to existing techniques – (3) tuning the backbone, (4) getting rid of the DINO-like contrastive loss. We provide evidence of all these design choices individually and collectively contributing to the performance gain in Table 6 of the Appendix.

---

> ### Author Response · Authors · 2024-11-19
> **[Q 2.1]**
>
> [Q 2.1] It is encouraged to clarify the novelty in the proposed UNIC, as its implementation looks very common and similar to some existing works.
> --
> The major novelty of this work is in the neighbor mining and cleaning strategies. We edit line 103 of the Introduction to emphasize this.
>
> Re-iterating our response to [W2.1] – According to the best of our knowledge of the literature, there isn’t much work that explores improving positives for this style of clustering and GCD. Utilizing negatives (instead of randomly sampling from the whole dataset) has not been explored in any prominent work.
>
> These ideas are explained in the Figure 02, Equation 3-4.
>
> Other than the overall implementation, we provide strong empirical evidence to justify our design choice on filtering with respect to second-order neighborhood sizes. To this end, Figure 4, Figure [8-11]. In addition, we provide an example visualization in the Appendix Figure 13.
>
> In addition, Appendix A.4 discusses the differences between our methodology from existing work. Furthermore, Appendix A.5 specifically answers the question about why our neighbor mining is novel.
>
> The authors would greatly appreciate it if the anonymous reviewer could point us to which literature overlaps with our design. In that case, we will be able to provide better differentiation between our work and the literature.

---

> ### Author Response · Authors · 2024-11-19
> **[Q 2.2]**
>
> [Q 2.2] The experiments are not comprehensive. It is needed to compare with other positive/negative mining strategies directly.
> --
> The authors have reported results from simpler mining strategies in Table 6 in the Appendix. The bottom row (7th row) of the table (90.80%) is obtained by our full mining strategy (which includes positive, cleaning, and negatives). The 6th row (89.04%) reports a simple strategy (which includes positives and negatives only). The 5th row (87.12%) reports the simplest strategy, which is to mine positives only.
>
> The authors would appreciate the anonymous reviewer feedback on whether these studies are enough or if we should compare them to something else specifically.

---

> ### Author Response · Authors · 2024-11-19
> **[Q 2.3]**
>
> [Q 2.3] Moreover, the experiments lack insightful analysis. For example, in Table 3, why the contrastive loss harms the performance of clustering;
> --
> Thank you for pointing out this. We have appended lines 409-410 to explain this scenario.

---

> ### Author Response · Authors · 2024-11-19
> **[Q 2.4]**
>
> [Q 2.4] For the GCD ablations in Table 4, it is curious why the labeled negative neighbors are inferior to the mined ones (when comparing the results in the second and third rows).
> --
> Thank you for pointing out this. We have appended lines 415-418 to explain this scenario. Another section has been appended to the Appendix (A.13) to elaborate on the underlying issue.

---

### Official Review · Reviewer_48qZ · 2024-11-06

**Soundness:** 3
**Presentation:** 2
**Contribution:** 3
**Rating:** 6
**Confidence:** 4

**Summary:**

This paper tackles the problem of image classification, tackling both the unsupervised clustering and partially supervised Generalised Category Discovery (GCD) problems. The proposed method is a simple one which first uses a pre-trained DINO backbone to mine positive and negative training samples in a dataset using (unsupervised) nearest neighbour search.

These positive and negative samples are then used to learn a parametric classifier (which provides a distribution over the ground truth number of classes in the dataset) with a positive and negative classification loss. The positive loss is the cross-entropy between model predictions on the anchor and those on the mined positives, while the negative loss is the inverse of this on the negative samples (though this is not clear to me).

The authors show state of the art results on both unsupervised clustering and GCD tasks on some standard benchmarks.

**Strengths:**

* The proposed method seems to be very simple and provide substantial gains over existing baselines in the GCD and unsupervised clustering literature.
* The simplicity of the approach means the solution can be readily extended to other partially supervised settings (e.g standard semi-supervised learning). The authors already demonstrate strong results on two popular tasks, but it is a positive that the method may find broader applicability.
* As far as I can see, the main hyper-parameters (size of the neighbourhood for the nearest neighbour mining, loss term weighting, steps in the mining) have been properly ablated.

**Weaknesses:**

* My main concern is over the formulation of the learning algorithm. Particularly, I find it difficult to understand Eq 6. The entropy here is computed between two scalar values rather than a distribution. Is this standard binary cross entropy / log loss? This would be more easily understood if written out in full, in my opinion (given that this is not a multi-class entropy problem).
* As mentioned in "Strengths", the method is simple and can be easily extended. Did the authors consider adding the positive/negative mining strategy directly to existing methods (e.g it might fit naturally on top of the GCD baseline).
* The authors could have evaluated on more datasets. For instance, it is common practise to evaluate on the full "SSB" suite (including Stanford Cars and FGVCAircraft) in GCD. The authors could also include a long-tail evaluation like Herbarium19, where the positive/negative mining strategy may behave differently.

Misc:
* I believe the presentation of this paper could be improved. e,g: Table 2 is small and difficult to read; Table 6 in uncentered, etc.

**Questions:**

* Did the authors consider adding the mining strategy on top of existing contrastive GCD methods (e.g the GCD baseline).
* Did the authors consider long-tail evaluations of their method?

---

> ### Author Response · Authors · 2024-11-19
> **Author response to Reviewer 48qZ**
>
> We thank the anonymous reviewer for their time and effort in producing a meaningful review. We will respond to the reviews that do not require further experimentation first. In the meantime, we will start running the additional experiments for incorporating UNIC Neighbor mining for the GCD baseline and testing the performance of UNIC on other datasets as suggested by the reviewer.
>
> We want to emphasize two minor suggestions to the summary.
>
> **[S1.1]** One of the major improvements of our method over existing work (which mostly uses simple nearest neighbor search for mining positives) is that we have a refining step that can greatly reduce the noise level of the positive neighbors. We emphasize this in Figure 2, Equation 4. In addition, further justification is given in Appendix A.3 and Figure 13.
>
> **[S1.2]** This is an attempt to clarify the utility of negative neighbors. Consider one reference image. Intuitively, we penalize the model for predicting the wrong clusters/classes. Existing work assumes every image except the reference image is of a different cluster than it (i.e. all other images are negatives). However, this assumption gives rise to many false negatives because there ought to be images of the same cluster as the reference. We propose our own neighbor mining algorithm to reduce these false negative neighbors. This is emphasized in Figure 2, Equation 3, Equation 6 (L_neg term). The effectiveness is shown in Figure 8 (mid), and Table 6.

---

> ### Author Response · Authors · 2024-11-19
> **[W 1.1]**
>
> [W 1.1] My main concern is over the formulation of the learning algorithm. Particularly, I find it difficult to understand Eq 6. The entropy here is computed between two scalar values rather than a distribution. Is this standard binary cross entropy / log loss? This would be more easily understood if written out in full, in my opinion (given that this is not a multi-class entropy problem).
> ---
> Thank you for pointing out the mistake. We corrected the wording to emphasize on a,b being scalars (probabilities) instead of being vectors (probability distributions). The edits are visible on lines 262 and 275.

---

> ### Author Response · Authors · 2024-11-19
> **[W 1.4]**
>
> [W 1.4]I believe the presentation of this paper could be improved. e,g: Table 2 is small and difficult to read; Table 6 in uncentered, etc.
> --
> Thank you for the suggestions. We increased the font size of Table 2 and centered Table 6. Please refer to the updated PDF.

---

> ### Author Response · Authors · 2024-12-03
> **[W 1.2]  and [W 1.3.1]**
>
> [W 1.2] As mentioned in "Strengths", the method is simple and can be easily extended. Did the authors consider adding the positive/negative mining strategy directly to existing methods (e.g it might fit naturally on top of the GCD baseline).
> --
>
> The authors thank the reviewer for this question, which enables us to justify some of our novelty – which goes beyond cleaning neighbors for existing pipelines.
>
>
> Firstly, almost every one of the GCD works in the literature treats the labeled set and the unlabelled set differently. The labeled set is trained on cross-entropy loss between the predicted class and ground truth class labels. The unlabelled data is used for training a contrastive component. There is no notion of neighbors in this work.
>
> The contributions of UNIC include, (1) getting rid of the cross-entropy loss that works directly using the ground truth class labels, (2) adding positive and negative neighbors for both labeled and unlabelled data, (3) cleaning positive neighbors, (4) multiple loss functions.
>
> Getting rid of the need for ground truth class labels explicitly in the training is the reason why UNIC can perform both GCD and image clustering. These labels are implicitly used for positive neighbors for old classes in the GCD case. However, the UNIC pipeline itself is agnostic to whether these labels are used or not.
>
> It is impossible to answer the question “How will UNIC’s neighbor mining improve the GCD baseline?” without significant changes to the GCD baseline code, which in turn would make the implementation very different from the GCD baseline proposed in their paper.
>
> The authors were unable to locate some work in GCD literature where adding our neighbor mining and cleaning was a natural extension.
>
> In contrast, most clustering literature has a notion of neighbors. Therefore, we have conducted experiments by fitting our mining strategies to existing works in a very natural fashion. The results and discussion are presented in A.4 FAQ(1) IS THIS BETTER THAN SCAN+DINO?
>
>
> [W 1.3.1]The authors could have evaluated on more datasets. For instance, it is common practise to evaluate on the full "SSB" suite (including Stanford Cars and FGVCAircraft) in GCD.
> --
>
> We thank the reviewer for the suggestion to include more experiments and strengthen our paper. We run experiments on CUB-200, StanfordCars-196, and Aircrafts-100 datasets. We run two experiments for each dataset. First, we experiment with the DINOv1 pre trained Vit-B/16 backbone similar to earlier GCD literature. Then, we follow experiments with SimGCD [ICCV 2023] pretrarined backbone similar to recent GCD literature SPTnet [ICLR 2024]. Please refer to the table below for experimental results.
>
> Our results show that UNIC suffers from the low-quality neighbors mined with DINOv1 backbone. However, UNIC outperforms the existing literature when it is able to utilize SimGCD pretrained backbones and achieve state-of-the-art.
> It should be noted that SPTnet achieves its improvement over the previous literature by adding additional prompt parameters to the ViT backbone. UNIC achieves better performance without changing the backbone architecture (Only tuning the last block).
>
>
> | **Algorithm**       | **Backbone**       | **CUB-200** (All/Old/New) | **Aircrafts-100** (All/Old/New) | **SCars-196** (All/Old/New) |
> |----------------------|--------------------|----------------------------|----------------------------------|-----------------------------|
> | kMeans (Vaze et al., 2022) | DINOv1          | 34.3 / 38.9 / 32.1         | 12.9 / 12.9 / 12.8              | 12.8 / 10.6 / 13.8          |
> | UNO+ (Fini et al., 2021)   | DINOv1          | 35.1 / 49.0 / 28.1         | 28.3 / 53.7 / 14.7              | 35.5 / 70.5 / 18.6          |
> | ORCA (Cao et al., 2021)    | DINOv1          | 36.3 / 43.8 / 32.6         | 31.6 / 32.0 / 31.4              | 31.9 / 42.2 / 26.9          |
> | GCD (Vaze et al., 2022)    | DINOv1          | 51.3 / 56.6 / 48.7         | 45.0 / 41.1 / 46.9              | 39.0 / 57.6 / 29.9          |
> | DCCL (Pu et al., 2023)     | DINOv1          | 63.5 / 60.8 / 64.9         | -- / -- / --                    | 43.1 / 55.7 / 36.2          |
> | Prompt CAL (Zhang et al., 2023) | DINOv1      | 62.9 / 64.4 / 62.1         | 52.2 / 52.2 / 52.3              | 50.2 / 70.1 / 40.6          |
> | UNIC                  | DINOv1          | 44.6 / 51.6 / 40.7      | 25.5 / 28.6 / 23.9           | -- / -- / --                |
> | SimGCD (Wen et al., 2023)   | DINOv1          | 60.3 / 65.6 / 57.7         | 54.2 / 59.1 / 51.8              | 53.8 / 71.9 / 45.0          |
> | | | | | |
> | **SPTnet (Wang et al., 2024)** | DINOv1+SimGCD | 65.8 / 68.8 / **65.1** | 59.3 / 61.8 / **58.1**   | 59.0 / 79.2 / **49.3** |
> | **UNIC (ours)**       | DINOv1+SimGCD    | **69.4** / **83.4** / 62.4 | **63.9** / **68.1** / 61.8 | **62.6** / **90.7** / 49.0 |

---

### Meta-Review · Area_Chair_friL · 2024-12-17

**Metareview:**

This paper proposes a method, called Utilizes Neighbor Information for Classification (UNIC) to solve both unsupervised and semi-supervised clustering settings.

The main strengths include: 1) simple yet substantial gains; 2) strong performance on two clustering tasks; 3) motivation is interesting; and 4) paper is well written.

The reviewers raised concerns and drawbacks of this paper. The authors have provided a rebuttal to solve them accordingly. After checking the rebuttal and comments from the other reviewers, the reviewers acknowledged that some of their concerns have been solved. However, several important concerns still remain, including: 1) the the technical contribution is limited; 2) lack of deep analysis of the improvement. Thus, three of the reviewers decided to kept their negative scores to this paper. On the other hand, the reviewer who gave positive score did not champion the acceptance to this paper. To this end, the AC thinks this paper cannot meet the requirement of ICLR at this point and thus regrets to recommend rejection.

**Additional Comments On Reviewer Discussion:**

The authors have provided a rebuttal and the reviewers have attended the discussion phase. However, the reviewers who gave negative scores think their concerns about the novelty and explanations of experiments are not well solved and thus kept the negative scores.

---

### Decision · Program_Chairs · 2025-01-22

Reject